# Directional and disruptive selection in populations structured by class and continuous ontogeny under incomplete plasticity

**Arthur Weyna** ⓘ*, **Charles Mullon, Laurent Lehmann**

Department of Ecology and Evolution, University of Lausanne, Lausanne, Switzerland

* arthur.weyna@unil.ch

## Abstract

Many organisms undergo ontogeny, whereby individuals change in state (e.g., in size, morphology, or condition) as they age. Understanding the evolution of traits influencing ontogeny is challenging because their fitness effects unfold across an individual's lifetime and may differ between classes such as sexes. Here, we analyse selection on non-plastic traits (e.g., fixed resource allocation strategies) that determine the development of dynamical states throughout life (e.g., body size), with consequences for fecundity and survival in age- and class-structured populations. Using invasion analysis, we derive expressions for directional and disruptive selection that decompose into age- and class-specific components. This allows us to identify convergence stable trait values, assess whether they are uninvadable or potentially experience evolutionary branching, and pinpoint the age and class pathways through which correlational and disruptive selection act. Applying our results to a model of growth under size-mediated sexual selection, we show how selection on growth rates distributes across ages and sexes, and how its relative strength depends on genetic correlations, individual ploidy and life-history. We also show how sex-specific developmental trade-offs and constraints can generate disruptive selection on male growth and favour the evolution of alternative male life histories. More broadly, our results highlight how adaptation is mediated by the interaction of development and demography, and provide tools to investigate how conflicts across ages and classes may influence senescence, sexual dimorphism, and the diversification of ontogenetic strategies.

## Author summary

How does natural selection shape traits that influence how organisms develop through their lifetime? This question is difficult because the effects of such traits unfold dynamically as individuals change in states such as size or condition, which in turn influence survival and reproduction throughout life. We formulate

**Data availability statement:** The authors confirm that all data underlying the findings are fully available without restriction. Mathematica notebooks that reproduce our results, and python scripts implementing our individual-based simulations are available at zenodo.org (https://doi.org/10.5281/zenodo.18303003).

**Funding:** The author(s) received no specific funding for this work.

**Competing interests:** The authors have declared that no competing interests exist.

a mathematical model that follows these causal links and characterises the gradual evolution of non-plastic traits affecting development. Our results apply to populations with different classes, such as males and females, that follow distinct developmental and demographic paths. Our model reveals when selection is stabilizing, holding populations fixed for an optimal trait value, and when it is disruptive, favouring trait diversification and the emergence of alternative life-history strategies. As an illustration, we analyse the evolution of sex-specific growth under size-based sexual selection. We show how the evolution of male and female growth rates depends on genetics, such as individual ploidy or genetic correlation between traits, but also on life history, which determines the relative strength of selection acting on trait expression at different ages. We further show how developmental trade-offs and demographic differences can generate disruptive selection on male growth, leading to the coexistence of fast-growing short-lived males and slow-growing long-lived ones. More broadly, our results link development, demography, and adaptation, and show how their interplay shapes biological diversity both within and between populations.

## 1 Introduction

Many traits expressed by an organism throughout its life influence not only its survival and reproduction, but also its ontogeny. The fitness effects of such traits can be immediate, when their expression acts on current reproduction, as well as delayed, when their expression acts on future reproduction by modulating current maintenance and growth. Understanding how selection acts on traits affecting ontogeny is thus difficult because their fitness effects unfold over an individual's entire lifetime. Additionally, these fitness effects often depend on the traits expressed by others and vary among classes of individuals, such as individuals of different sexes or ploidies, in different environments, or in different conditions. For instance, growth allocation can lead to different outcomes in males and females of the same population due to sex-specific developmental trajectories and sexual selection (e.g., more individuals of a given size can increase competition and sex-specific size variation can influence mate choice). Such interactions between ontogeny, class structure and social effects make it challenging to understand how selection shapes traits affecting ontogeny.

The inheritance and expression of traits affecting ontogeny can be conceptualised in different ways, with important implications for the operation of natural selection. Consider for example the growth allocation schedule of females and males in some vertebrate species. At one extreme, this schedule could be a completely plastic trait, with allocation varying independently across age and sex. This perspective, often adopted in evolutionary life-history models (e.g., [1–3]), allows trait expression to be adjusted optimally to each context, albeit within preset physiological constraints (e.g., physical boundaries in trait space, available resources). Such complete plasticity, however, implies the existence of a number of genetic loci large enough that expression can be matched to each possible context, with mutations able to generate

age- and/or class-specific changes. It also implies that selection must be strong enough, on these numerous loci, to perfectly fine-tune trait expression. This contrasts with the fact that plasticity is often incomplete owing to genetic constraints (i.e., due to trait expression being controlled by a limited number of loci), leading to similar trait values being expressed across contexts. Under such constraints, selection acts on a shared genetic architecture, potentially generating antagonistic pressures across age [4] or classes [5]. In humans for instance, some mutations in BRCA genes are associated with increased mortality in older individuals (via higher rates of breast and ovarian cancer), but also with increased fertility in young individuals (via earlier age at paternity in men and lower miscarriage rates in women; [6]).

Mathematical models have been useful for analysing selection on traits affecting ontogeny under complete and incomplete plasticity. These models typically consider gradual trait evolution, whereby under a constant influx of mutations with small effects on traits, a population evolves under directional selection towards a local evolutionary equilibrium. Under complete age-wise plasticity, directional selection on age-specific trait expression depends on its effect on current, as well as future reproduction, via changes in development or survival (e.g., [2,7–10]). These models show that by treating ontogeny as a dynamical system at the individual level, it becomes possible to track the future fitness consequences of age-specific trait expression, such that each instant of expression can be optimized. More recent studies have applied the same future-accounting methods to cases where plasticity is incomplete [9,11]. When a shared genetic architecture forces the same trait value to be expressed at different ages or times, directional selection can be understood as a weighted sum of fitness effects over all instants where a trait value is expressed [9,11,12]. The weights in that sum then reflect the discount in selection efficiency that applies to later trait expression, due to the thinning of cohorts and to the reduction in remaining reproduction opportunities [13]. This body of theory provides a way to analyse the outcome of directional selection on traits that affect ontogeny under genetic constraints, while also accounting for age-specific demography and social interactions.

Once a population has converged towards a local evolutionary equilibrium under directional selection (a "convergence stable trait" value, [14,15]), it may either remain genetically monomorphic due to stabilising selection or become polymorphic owing to disruptive selection (e.g., [15–17]). However, disruptive selection on traits that affect ontogeny remains understudied [12]. Nevertheless, empirical evidence suggests that selection plays a role in shaping life-history polymorphisms. In the Atlantic salmon for instance, young males can adopt alternative ontogenetic trajectories that favour either short-term mating or delayed reproduction after migration and growth [18]. This polymorphism is in part explained by genetic variation in developmental thresholds, but current theory offers no general method to assess whether this variation is favoured by disruptive selection. A second limitation is that most current analyses of selection on traits affecting ontogeny consider populations structured by age only. Yet many natural populations are also structured in various discrete classes (e.g., sexes, ploidies or local environments) in which the consequences of trait expression differ. Polymorphisms in such populations can occur within a single class [18] or span multiple classes, sometimes with overlapping genetic control [19]. It is unknown to what extent genetic constraints, by coupling selection pressures acting in different classes, may affect life history evolution and diversification.

Here, we fill these gaps by extending current theory to analyse directional and disruptive selection on a finite number of quantitative traits expressed constantly throughout life history (i.e., incomplete plasticity) and that affect ontogeny in class-structured populations. We consider selection on traits expressed by individuals belonging to different classes with specific development, ploidy, demography, and social context. We derive expressions for both directional and disruptive selection on these traits, that we partition into class- and age-wise components, and provide numerical tools that facilitate the calculation of these expressions. Additionally, we use these results to study a model of size-mediated sexual selection under a sex-specific survival-growth trade-off and frequency-dependent competition. In this application, we find that under strong sexual selection, disruptive selection on male growth rates leads to the emergence of multiple male life histories: some grow rapidly to large size, thereby obtaining mating opportunities with highly fecund females but suffering higher mortality; others remain small, specialise in less fecund mates, and survive longer. Beyond this specific example,

our results allow for analyses of directional and disruptive selection under a wide range of evolutionary contexts including ecological and social feedbacks, and thus open the door to a sharper understanding of the selective forces that shape life history and its variation within populations.

## 2 Methods

### 2.1 Biological assumptions

We consider a large age-structured population in continuous time, under constant global abiotic environmental conditions. Each individual in the population belongs to one of a finite number $n_c$ of classes, such as sex, mating type, or habitat state. The class of an individual is determined at birth and does not change during its lifetime. We leave unspecified at this point the mode of reproduction and ploidy of each class, but assume that they are constant properties of the population. To implement class-specific ontogeny, we associate to a focal individual of class $j$ and age $a$ a vector $\boldsymbol{x}_j(a) = (x_{j,1}(a), ..., x_{j,n_x}(a))$ of $n_x$ class-specific internal state variables, where $x_{j,m}(a)$ is a scalar giving the value of the focal's $m$-th internal state at age $a$ (see Table 1 for a summary of the symbols used throughout), and where $n_x$ is assumed to be independent of age. Internal states can represent any relevant biological quantitative feature that changes with age, such as the mass or size of a given organ, or the probability of occurrence of a physiological event (e.g., moulting or menopause).

We assume that the rates of change $\mathrm{d}\boldsymbol{x}_j(a)/\mathrm{d}a$ in internal states in a focal individual of age $a$ and class $j$ depend deterministically on its internal states and on a vector $\boldsymbol{z}_j = (z_{j,1}, ..., z_{j,n_z})$ of $n_z$ class-specific evolving quantitative traits, where $z_{j,l}$ is a scalar giving the value of the $l$-th trait expressed in class $j$ (called trait $(j, l)$ hereafter). Trait values are assumed to be constant properties of individuals throughout ontogeny (e.g., constant allocation proportions), fully determined by their genetic value (no stochastic environmental effects on trait expression) under additive genetic action (no dominance). All traits are assumed to be encoded by the same locus under a continuum-of-alleles model without mutational bias (e.g., [20,21]).

Besides their effect on development, the internal states and traits of an individual of class $j$ are assumed to also influence its vital rates, that is its mortality rate $\mu_j$, and the rate $f_{ij}$ at which it produces offspring of class $i$. Hereafter, we will refer to those quantities conditional on both a class $j$ parent and a class $i$ offspring as *class-oriented* quantities (e.g., $f_{ij}$ is the class-oriented fecundity rate). An individual's developmental rates and vital rates are further assumed to depend deterministically on a set of environmental variables, that are common to all individuals in the population but may affect individuals differently depending on their internal states. These external environmental variables can be demographic (e.g., population density) or ecological (e.g., available resources, pathogen density), and their dynamics can thus depend on the population.

Taken together, these assumptions let us derive the evolutionary dynamics of the evolving traits in populations structured by class and ontogeny. Our objective is to establish the first and second order conditions necessary to classify the singular points of gradual evolutionary dynamics [17]. To this aim, it is sufficient to carry out an invasion analysis (e.g., [12,17,22,23,24,25]), that is to focus on the invasion process of a rare mutant allele inducing a trait change in a population fixed for some resident allele (i.e., monomorphic for some resident trait).

### 2.2 The mutant basic reproductive number

To account for the effect of mutant alleles on mutant trait expression under arbitrary ploidy and reproductive systems, we denote by $z_{j,l}(u_{j,l}, v_{j,l})$ the value of trait $(j, l)$ in a mutant individual (i.e., an individual carrying at least one mutant allele), where $u_{j,l}$ is the mutant allelic value specific to trait $(j, l)$ and $v_{j,l}$ is the corresponding resident allelic value. The allelic value of an allele is defined here as the trait value induced by that allele in an individual carrying only copies of that allele. This construction allows to cover simple systems such as, for instance, haploidy (where mutants carry a single mutant

**Table 1. List of symbols used in the main text.**

| Symbol | Meaning |
|---|---|
| **Parameters** | |
| $n_c$ | Number of classes |
| $n_z$ | Number of evolving traits in each class |
| $n_x$ | Number of internal states |
| $\gamma_{ij}$ | Expected number of alleles transmitted to offspring of class $i$ by a parent of class $j$ |
| $c_{ij}$ | Proportion of individuals of class $i$ within the offspring of individuals of class $j$ |
| **Variables** | |
| $u_{j,l}, v_{j,l}$ | Allelic value for trait $l$ in class $j$ encoded by mutant and resident alleles, respectively |
| $\boldsymbol{u}_j, \boldsymbol{v}_j$ | Vectors of mutant and resident allelic values in class $j$, respectively |
| $\boldsymbol{u}, \boldsymbol{v}$ | Vectors collecting mutant and resident allelic values across all classes ($\boldsymbol{v}$ is also used as a shorthand for resident traits as $\boldsymbol{z}(\boldsymbol{v}, \boldsymbol{v}) = \boldsymbol{v}$) |
| $z_{j,l}(u_{j,l}, v_{j,l})$ | Value of trait $l$ in a mutant of class $j$ |
| $\boldsymbol{z}_j(\boldsymbol{u}_j, \boldsymbol{v}_j)$ | Vector of trait values in a mutant of class $j$ |
| $\boldsymbol{z}(\boldsymbol{u}, \boldsymbol{v})$ | Vector collecting mutant trait values across all classes |
| **Functions** | |
| $l_j(a), l_j^\circ(a)$ | Survival probability (survivorship) to age $a$ of a mutant and resident individual of class $j$, respectively |
| $\boldsymbol{x}_j(a), \boldsymbol{x}_j^\circ(a)$ | Internal state value at age $a$ of a mutant and resident individual of class $j$, respectively |
| $f_{ij}(\boldsymbol{z}_j, \boldsymbol{x}_j(a), \boldsymbol{v})$ | Fecundity rate: rate of production of offspring of class $i$ by a mutant of class $j$ at age $a$ |
| $f_j(\boldsymbol{z}_j, \boldsymbol{x}_j(a), \boldsymbol{v})$ | Total fecundity rate: total rate of production of offspring by a mutant of class $j$ at age $a$ |
| $\mu_j(\boldsymbol{z}_j, \boldsymbol{x}_j(a), \boldsymbol{v})$ | Mortality rate in a mutant of class $j$ and age $a$ |
| $\boldsymbol{g}_j(\boldsymbol{z}_j, \boldsymbol{x}_j(a), \boldsymbol{v})$ | Vector of rates of change in internal states at age $a$ in a mutant of class $j$ |
| $\mathbf{R}(\boldsymbol{u}, \boldsymbol{v})$ | Mutant next-generation matrix |
| $R_0(\boldsymbol{u}, \boldsymbol{v})$ | Mutant basic reproductive number |
| $\boldsymbol{\nu}^\circ, \nu_i^\circ$ | Leading left-eigenvector of $\mathbf{R}^\circ = \mathbf{R}(\boldsymbol{v}, \boldsymbol{v})$ and its element $i$, i.e., resident allelic reproductive values at birth of class $i$ |
| $\boldsymbol{q}^\circ, q_j^\circ$ | Leading right-eigenvector of $\mathbf{R}(\boldsymbol{v}, \boldsymbol{v})$ and its element $j$, i.e., the resident allelic frequency at birth of class $i$ |
| $R_{ij}(\boldsymbol{u}_j, \boldsymbol{v})$ | Expected lifetime fecundity via class-$i$ offspring of a mutant of class $j$ |
| $R_j(\boldsymbol{u}_j, \boldsymbol{v})$ | Total expected lifetime fecundity of a mutant of class $j$ |
| $H_{ij}(\boldsymbol{z}_j, l_j(a), \boldsymbol{x}_j(a), \boldsymbol{v})$ | Class-oriented Hamiltonian |
| $\lambda_{ij}^l(a), \boldsymbol{\lambda}_{ij}^x(a)$ | Class-oriented costates associated to survivorship and internal states |
| $H_j(\boldsymbol{z}_j, l_j(a), \boldsymbol{x}_j(a), \boldsymbol{v})$ | Class-specific Hamiltonian |
| $\lambda_j^l(a), \boldsymbol{\lambda}_j^x(a)$ | Class-specific costates associated to survivorship and internal states, respectively |
| $\boldsymbol{s}_j(\boldsymbol{v})$ | Selection gradient specific to class $j$ |
| $s_{j,l}(\boldsymbol{v}), \hat{s}_{j,l}(a, \boldsymbol{v})$ | Total and age-specific directional selection on trait $(j, l)$ |
| $\mathbf{J}(\boldsymbol{v}), \mathbf{J}_{ij}(\boldsymbol{v})$ | Jacobian matrix and its block $(i, j)$ |
| $\mathbf{H}(\boldsymbol{v}), \mathbf{H}_{ij}(\boldsymbol{v})$ | Hessian matrix and its block $(i, j)$ |
| $h_{ij,kl}(\boldsymbol{v}), \hat{h}_{ij,kl}(a, \boldsymbol{v})$ | Total and age-specific quadratic selection on traits $(i, k)$ and $(j, l)$ |

allele, $z_{j,l}(u_{j,l}, v_{j,l}) = u_{j,l}$), diploidy under full genetic mixing and dosage compensation (i.e., where all mutant individuals are heterozygotes with trait $z_{j,l}(u_{j,l}, v_{j,l}) = \frac{1}{2}(u_{j,l} + v_{j,l})$) or any combination of both across classes (e.g., haplodiploidy). This notation for trait $z_{j,l}$ also accounts for more complicated cases, such as systems with polyploid classes, or where mutants can carry more than one mutant allele (e.g., diploid mutants produced via selfing; see for instance [26]). With this, we let $\mathbf{z}(\mathbf{u}, \mathbf{v}) = (\mathbf{z}_1(\mathbf{u}_1, \mathbf{v}_1), ..., \mathbf{z}_{n_c}(\mathbf{u}_{n_c}, \mathbf{v}_{n_c}))$ stand for the vector collecting mutant trait values in each class, where $\mathbf{z}_j(\mathbf{u}_j, \mathbf{v}_j) = (z_{j,1}(u_{j,1}, v_{j,1}), ..., z_{j,n_z}(u_{j,n_z}, v_{j,n_z}))$ is the vector of traits expressed by a mutant individual of class $j$. Therein, $\mathbf{u} = (\mathbf{u}_1, ..., \mathbf{u}_{n_c})$, where $\mathbf{u}_j = (u_{j,1}, ..., u_{j,n_z})$ stands for the vector of mutant allelic values specific to class $j$, and likewise $\mathbf{v} = (\mathbf{v}_1, ..., \mathbf{v}_{n_c})$, where $\mathbf{v}_j = (v_{j,1}, ..., v_{j,n_z})$ stands for the vector of resident allelic values specific to class $j$. Under the assumptions of additive gene action and dosage compensation, individuals carrying only resident alleles have traits equal to the resident allelic value. Thus, resident trait $(j, l)$ can conveniently be written as $z^{\circ}_{j,l} = z_{j,l}(v_{j,l}, v_{j,l}) = v_{j,l}$ (here and hereafter, we use the circle $\circ$ superscript to denote evaluation at resident trait values, i.e., $u_{j,l} = v_{j,l}$), such that $\mathbf{z}^{\circ} = \mathbf{v}$.

Having defined trait values in terms of allelic values, we now turn to the conditions of spread of a mutant allele. It follows from our demographic assumptions, and from standard results on multi-type age-dependent branching processes [27], that the fate of a lineage of mutant allelic copies encoding allelic values $\mathbf{u}$, and descending from a single copy introduced into a resident population monomorphic for allelic values $\mathbf{v}$ at equilibrium (i.e., where environmental processes have reached a fixed attractor; e.g., [23,28,29]), can be predicted from its basic reproductive number $R_0(\mathbf{u}, \mathbf{v})$. This is the expected number of descendant mutant copies produced over its whole lifetime by an individual carrying the mutant allele, averaged over all possible individual contexts in which the mutant allele can be found in. Specifically, a mutant lineage starting from a single copy goes extinct with certainty when $R_0(\mathbf{u}, \mathbf{v}) \leq 1$, but has a positive probability of not going extinct when $R_0(\mathbf{u}, \mathbf{v}) > 1$.

The basic reproductive number is obtained as the leading eigenvalue of the mutant next-generation matrix $\mathbf{R}(\mathbf{u}, \mathbf{v})$, whose $ij$-th element is the expected number of descendant copies that a mutant allele found in an individual of class $j$ produces over its lifetime, via descendant individuals of class $i$, when the population is monomorphic for $\mathbf{v}$ ([27,30,31]; see appendix A.1 in S1 Text for details). This element can be written as the product $\gamma_{ij} R_{ij}(\mathbf{u}_j, \mathbf{v})$, where $R_{ij}(\mathbf{u}_j, \mathbf{v})$ is the expected lifetime number of offspring of class $i$ produced by a focal mutant individual of class $j$ (i.e., individual class-oriented lifetime fecundity), and where $\gamma_{ij}$ gives the expected number of mutant allelic copies transmitted to one offspring of class $i$ by a parent of class $j$ (i.e., converting *individual* fecundity into *allelic* fecundity and assumed to be constant). In a well-mixed diploid population with sexual reproduction for instance, $\gamma_{ij} = 1/2$ for any $i$ and $j$ as diploid mutants carry one mutant allele that is transmitted to half their offspring.

We express individual class-oriented lifetime fecundity as

$$R_{ij}(\mathbf{u}_j, \mathbf{v}) = \int_0^\infty l_j(a) f_{ij}(\mathbf{z}_j(\mathbf{u}_j, \mathbf{v}_j), \mathbf{x}_j(a), \mathbf{v}) \, \mathrm{d}a,$$

(1)

that is as the integral over all possible ages of the product of the probability $l_j(a)$ to survive to age $a$ (i.e., survivorship) with $f_{ij}(\mathbf{z}_j(\mathbf{u}_j, \mathbf{v}_j), \mathbf{x}_j(a), \mathbf{v})$, which is the rate at which a mutant individual of age $a$ and class $j$ produces offspring of class $i$ and depends on the individual's state $\mathbf{x}_j(a)$. Survivorship and state values themselves constitute the unique solution to the system of autonomous ordinary differential equations (ODEs):

$$\frac{\mathrm{d}l_j(a)}{\mathrm{d}a} = -\mu_j(\mathbf{z}_j(\mathbf{u}_j, \mathbf{v}_j), \mathbf{x}_j(a), \mathbf{v}) \, l_j(a) \quad \text{with i.c.} \quad l_j(0) = 1,$$

$$\frac{\mathrm{d}\mathbf{x}_j(a)}{\mathrm{d}a} = \mathbf{g}_j(\mathbf{z}_j(\mathbf{u}_j, \mathbf{v}_j), \mathbf{x}_j(a), \mathbf{v}) \quad \text{with i.c.} \quad \mathbf{x}_j(0) = \mathbf{x}_{j,0},$$

(2)

where $\mu_j(z_j(u_j, v_j), x_j(a), v)$ and $g_j(z_j(u_j, v_j), x_j(a), v)$ are, respectively, the mortality rate and vector of developmental rates of a mutant of class $j$ (with $x_{j,0}$ being a fixed vector of initial values for the internal states).

The fact that the fecundity, mortality, and developmental rates in eqs (1) and (2) do not depend explicitly on age $a$ reflects the modelling assumption that all environmental and physiological effects on individuals are mediated by their internal state variables. The biological rationale behind this assumption is that organisms typically respond not to time per se, but to internal cues (e.g., energy reserves, physiological state, accumulated damage) or to external cues (e.g., temperature, photoperiod) that may correlate with age or other aspects of condition. Under this rationale, the dependence of eqs (1) and (2) on the resident traits $v$ summarizes all interactions that a focal mutant has with its environment, when this environment is at an equilibrium (which is assumed throughout and entails no global environmental fluctuations). This is because all demographic and/or ecological state variables are then fully determined by the resident population, and thus by resident traits. In other words, environmental processes, as complex as they may be, do not need to be modelled explicitly for a general characterization of selection in our model. Within specific applications however, the environment does need to be modelled, such that the effect of $v$ on a mutant can be measured (see for instance [32] for ecological feedbacks, or [11] for behavioral feedbacks. See also the example model of section 4 where simple density-dependent fecundity is implemented).

In summary, our model decomposes class-oriented lifetime reproductive success in terms of three embedded phenotypic layers. The highest layer is composed of vital rates (i.e., fecundity and mortality rates), which depend on individual traits, current states and the environment. The second layer is that of the states, which summarize the current behavioural, physiological or morphological states of individuals resulting from past trait expression and past interactions with their environment (since the rates of change in states depend on traits and on the environment). The third layer is that of evolving traits, which depend only on allelic values and summarize how these values are aggregated and expressed by individuals. In turn, allelic values are encoded by single alleles, and the set of alleles carried by individuals constitute their genotype. Because we concentrate on invasion analyses, we need not consider more than two alleles at a time, a mutant and a resident allele.

## 2.3 Local analysis and evolutionary dynamics

### 2.3.1 Directional and quadratic selection gradients.
We now assume weak mutation effects and write the mutant's allelic value as a small perturbation $u_{j,l} = v_{j,l} + \epsilon \eta_{j,l}$ of the resident's allelic value, where $\eta_{j,l}$ is a trait-specific random mutation effect and $\epsilon \ll 1$ is a small parameter (assuming that mutations are such that $u_{j,l}$ always lies within the domain of $z_{j,l}$, for any trait $l$ in any class $j$). The basic reproductive number of a mutant allele can then be Taylor expanded around $\epsilon = 0$ as

$$R_0(u, v) = 1 + \sum_{j=1}^{n_c} \eta_j \cdot s_j(v)\epsilon + \frac{1}{2} \sum_{i=1}^{n_c} \sum_{j=1}^{n_c} \eta_i^\mathsf{T} H_{ij}(v)\eta_j \epsilon^2 + O(\epsilon^3),$$

(3a)

where we used the fact that in a resident population at demographic equilibrium, $R_0(v, v) = 1$, and $O(\epsilon^3)$ is a remainder of order $\epsilon^3$. Here, $\eta_j = (\eta_j, ..., \eta_{j,n_z})$ is the effect of the mutant in class $j$; $s_j(v)$ is a $n_z \times 1$ vector of directional selection coefficients, with element $l$ given by

$$s_{j,l}(v) = \left. \frac{\partial R_0(u, v)}{\partial u_{j,l}} \right|_{u=v} = \left. \frac{\partial R_0(u, v)}{\partial z_{j,l}} \right|_{z=v} \times \left. \frac{\partial z_{j,l}(u_{j,l}, v_{j,l})}{\partial u_{j,l}} \right|_{u=v};$$

(3b)

$H_{ij}(v)$ is a $n_z \times n_z$ matrix of quadratic selection coefficients with $(k, l)$-element

$$h_{ij,kl}(\boldsymbol{v}) = \left.\frac{\partial^2 R_0(\boldsymbol{u}, \boldsymbol{v})}{\partial u_{i,k} \partial u_{j,l}}\right|_{\boldsymbol{u}=\boldsymbol{v}} = \left.\frac{\partial^2 R_0(\boldsymbol{u}, \boldsymbol{v})}{\partial z_{i,k} \partial z_{j,l}}\right|_{\boldsymbol{z}=\boldsymbol{v}} \times \left.\frac{\partial z_{i,k}(u_{i,k}, v_{i,k})}{\partial u_{i,k}}\right|_{\boldsymbol{u}=\boldsymbol{v}} \times \left.\frac{\partial z_{j,l}(u_{j,l}, v_{j,l})}{\partial u_{j,l}}\right|_{\boldsymbol{u}=\boldsymbol{v}},$$

(3c)

where we used the facts that $R_0(\boldsymbol{u}, \boldsymbol{v})$ implicitly depends on $\boldsymbol{z}(\boldsymbol{u}, \boldsymbol{v})$ and that $\boldsymbol{z}^\circ = \boldsymbol{v}$. Finally, '·' denotes the dot product, and '⊤' denotes transposition in eq (3a).

The quantity $s_{j,l}(\boldsymbol{v})$ (eq 3b) is the effect of a marginal change in the mutant allelic value $u_{j,l}$ on the mutant's reproductive number. It can be decomposed as the product of two components (right-hand side of eq 3b): the marginal effect of $z_{j,l}$ on $R_0(\boldsymbol{u}, \boldsymbol{v})$, and the effect of a change in allelic value on trait expression. Because we assume additive gene action, this second term is a constant independent of trait values for a given class $j$: it gives the relative genetic contribution of a mutant allele to the individual's trait value (e.g., one-half under diploidy or one under haploidy). If $s_{j,l}(\boldsymbol{v}) > 0$, selection favours an increase in that trait; if $s_{j,l}(\boldsymbol{v}) < 0$, it favours a decrease (see Table 2 for a summary of the types of selection considered throughout).

Meanwhile, the quantity $h_{ij,kl}(\boldsymbol{v})$ (eq 3c) is proportional to the multiplicative effects of joint changes in $z_{i,k}$ and $z_{j,l}$ on the mutant's reproductive number, beyond their individual effects. When $i \neq j$ or $k \neq l$, it quantifies how a change in the first trait modulates the effect of a change in the second. In other words, $h_{ij,kl}(\boldsymbol{v})$ describes correlational selection on traits $(i, k)$ and $(j, l)$: positive values indicate that $R_0(\boldsymbol{u}, \boldsymbol{v})$ increases faster with one trait when the other is larger – meaning that quadratic selective effects favour similar changes in the two traits – whereas negative values indicate that $R_0(\boldsymbol{u}, \boldsymbol{v})$ increases faster with one trait when the other is smaller, favouring opposite changes. For $i = j$ and $k = l$, a negative value of $h_{ii,kk}(\boldsymbol{v})$ indicates that $R_0(\boldsymbol{u}, \boldsymbol{v})$ decelerates with increasing deviation from the resident trait along a single trait axis $(i, k)$, while a positive value of $h_{ii,kk}(\boldsymbol{v})$ indicates that $R_0(\boldsymbol{u}, \boldsymbol{v})$ accelerates along that axis – meaning that quadratic selection favours trait values close or away from the current resident value, respectively.

**Table 2. Summary of selection types.**

| Selection | Measure | Effect |
|---|---|---|
| Directional | $s_{j,l}(\boldsymbol{v})$ | Favours an increase ($s_{j,l}(\boldsymbol{v}) > 0$) or decrease ($s_{j,l}(\boldsymbol{v}) < 0$) in the genetic value for trait $(j, l)$. These values are collected in the selection gradient vector $\boldsymbol{s}_j(\boldsymbol{v})$, whose $l$-th element is $s_{j,l}(\boldsymbol{v})$ and gives the directional selection in class $j$. |
| Quadratic | $h_{ij,kl}(\boldsymbol{v}^*)$ | Favours an increase ($h_{ij,kl}(\boldsymbol{v}^*) > 0$) or decrease ($h_{ij,kl}(\boldsymbol{v}^*) < 0$) in the (co)variance between the genetic values for traits $(i, k)$ and $(j, l)$ at a singular strategy $\boldsymbol{v}^*$. These values are collected in the Hessian blocks $\mathbf{H}_{ij}(\boldsymbol{v}^*)$, whose $(k, l)$-element is $h_{ij,kl}(\boldsymbol{v}^*)$. The full Hessian matrix $\mathbf{H}(\boldsymbol{v}^*)$ consists of all such blocks. |
| Correlational | $h_{ij,kl}(\boldsymbol{v}^*)$, with $i \neq j$ or $k \neq l$ | Favours an increase ($h_{ij,kl}(\boldsymbol{v}^*) > 0$) or decrease ($h_{ij,kl}(\boldsymbol{v}^*) < 0$) in the genetic covariance between different traits $(i, k)$ and $(j, l)$. This captures correlational selection across classes (if $i \neq j$) or between different traits within a class (if $i = j$ and $k \neq l$). |
| Stabilising | $h_{jj,ll}(\boldsymbol{v}^*) < 0$ | Favours a decrease in the genetic variance of trait $(j, l)$. |
| Disruptive | $h_{jj,ll}(\boldsymbol{v}^*) > 0$ | Favours an increase in the genetic variance of trait $(j, l)$. |
| Net stabilising | $\mathrm{eig}\left(\mathbf{H}(\boldsymbol{v}^*)\right) < 0$ | Disfavours any mutant away from $\boldsymbol{v}^*$ when all traits jointly evolve. $\mathrm{eig}\left(\mathbf{H}(\boldsymbol{v}^*)\right)$ refers to the leading eigenvalue of $\mathbf{H}(\boldsymbol{v}^*)$. |
| Net disruptive | $\mathrm{eig}\left(\mathbf{H}(\boldsymbol{v}^*)\right) > 0$ | Favours divergence away from $\boldsymbol{v}^*$ when all traits evolve jointly. Divergence occurs along the eigenvector associated with the leading eigenvalue of $\mathbf{H}(\boldsymbol{v}^*)$. |

Note that a singular strategy $\boldsymbol{v}^*$ is defined here as a vector of trait values for which directional selection vanishes (see eq 5 for independent traits, and eq 21 for genetically correlated traits).

**2.3.2 Mutation limited evolutionary dynamics.** Directional and quadratic selection coefficients inform gradual trait evolution when mutations are rare and have weak effects on the traits. Specifically, when mutations have small effects (i.e., $\epsilon \ll 1$), the 'invasion implies substitution' principle for class-structured populations [33] states that a mutant will increase (respectively, decrease) in frequency and eventually replace the resident (be lost) whenever $\sum_{j=1}^{n_c} \eta_j \cdot \mathbf{s}_j(\boldsymbol{v})$ is positive (respectively, negative). This implies the existence of a mutation-limited regime in which evolution proceeds gradually, as a sequence of trait substitutions along the direction favoured by selection. This sequence halts only when the population reaches a phenotypic point $\boldsymbol{v}$ where

$$\sum_{j=1}^{n_c} \eta_j \cdot \mathbf{s}_j(\boldsymbol{v}) \leq 0 \tag{4}$$

for all mutations $\eta = (\eta_1, ..., \eta_{n_c})$ in a close neighbourhood of $\boldsymbol{v}$. When the trait space is bounded and $\boldsymbol{v}$ is at the boundary, new mutations can only move traits inward or along the boundary. Condition (4) can then be satisfied if directional selection opposes all admissible mutations (i.e., or equivalently if selection favours trait values outside of the authorized range, such that corresponding elements of $\eta_j$ and $\mathbf{s}_j(\boldsymbol{v})$ always have opposing sign). In what follows, we focus on cases where condition (4) is met in the interior of trait space.

When mutation effects on trait values are not exactly equal across traits – that is when the distribution of mutation effect is not confined to a lower-dimensional subspace of trait space (e.g., $\eta$ is sampled from a multi-normal distribution over $n_c \times n_z$ with full-rank covariance matrix) – there always exists some mutation that affects one trait without changing the others. Therefore, on the long term and unless mutation effects are fully correlated across traits, traits can be thought of as being unconstrained genetically. In this case, condition (4) is satisfied in the interior of trait space if, for some trait value vector $\boldsymbol{v}^*$,

$$s_{j,l}(\boldsymbol{v}^*) = 0 \quad \text{for each trait } (j, l). \tag{5}$$

Any $\boldsymbol{v}^*$ satisfying eq (5) will be called a singular trait value and constitutes a candidate evolutionary equilibrium. But will evolution converge to such a singular trait?

A sufficient condition for $\boldsymbol{v}^*$ to act as a local attractor of the evolutionary dynamics – i.e., for the population to evolve toward $\boldsymbol{v}^*$ via a trait substitution sequence – is that the Jacobian matrix $\mathbf{J}(\boldsymbol{v}^*)$ is negative definite ([34, p. 189]). This matrix is a $n_c \times n_c$ block matrix whose block $(i, j)$ is a matrix $\mathbf{J}_{ij}(\boldsymbol{v}^*)$ with element $(k, l)$ given by

$$j_{ij,kl} = \left. \frac{\partial s_{i,k}(\boldsymbol{v})}{\partial v_{j,l}} \right|_{\boldsymbol{v}=\boldsymbol{v}^*}. \tag{6}$$

The matrix $\mathbf{J}(\boldsymbol{v}^*)$ is negative definite if the leading eigenvalue of its symmetric part, $(\mathbf{J}(\boldsymbol{v}^*) + \mathbf{J}^{\mathsf{T}}(\boldsymbol{v}^*))/2$, is negative. In that case, $\boldsymbol{v}^*$ is said to be strongly convergence stable (*sensu* [34]). The term 'strongly' reflects that convergence occurs from all nearby trait values, regardless of the mutational covariance structure. In contrast, when $\mathbf{J}(\boldsymbol{v}^*)$ is not negative semi-definite, there is some mutational covariance structure for which $\boldsymbol{v}^*$ is a repellor of the evolutionary dynamics ([34, p. 198]).

By eq (3a), if the population has reached a singular trait value $\boldsymbol{v}^*$, then a sufficient condition for local uninvadability, i.e., for any nearby mutant to be selected against and thus for selection to be stabilising at $\boldsymbol{v}^*$, is that

$$\sum_{i=1}^{n_c} \sum_{j=1}^{n_c} \eta_i^{\mathsf{T}} \mathbf{H}_{ij}(\boldsymbol{v}^*) \eta_j < 0 \quad \text{for all mutations } \eta = (\eta_1, ..., \eta_{n_c}). \tag{7}$$

Eq (7) holds if and only if the block matrix $\mathbf{H}(\boldsymbol{v}^*)$, whose block $(i, j)$ is $\mathbf{H}_{ij}(\boldsymbol{v}^*)$, is negative definite (i.e., $\boldsymbol{\eta}^\mathsf{T}\mathbf{H}(\boldsymbol{v}^*)\boldsymbol{\eta} < 0$ for all $\boldsymbol{\eta}$). Since $\mathbf{H}(\boldsymbol{v}^*)$ is a symmetric real matrix, it is negative definite if and only if all its eigenvalues are negative.

When $\mathbf{H}(\boldsymbol{v}^*)$ has at least one positive eigenvalue (i.e., $\mathbf{H}(\boldsymbol{v}^*)$ is not negative semi-definite), there exists a set of mutation directions – specifically those aligned with the leading eigenvector of $\mathbf{H}(\boldsymbol{v}^*)$ – that are favoured on both sides of the singular strategy $\boldsymbol{v}^*$. In other words, selection is disruptive at $\boldsymbol{v}^*$: it favours divergence away from the resident trait value. A sufficient condition for $\mathbf{H}(\boldsymbol{v}^*)$ to not be negative semi-definite is that any diagonal block $\mathbf{H}_{jj}(\boldsymbol{v}^*)$ is not negative semi-definite (since diagonal blocks are symmetric matrices, [35, Theorem 2.2, p. 1062]). In turn, a diagonal block $\mathbf{H}_{jj}(\boldsymbol{v}^*)$ is not negative semi-definite if any of its diagonal elements is positive, i.e., if $h_{jj,kk}(\boldsymbol{v}^*) > 0$ for some trait $(j, k)$. At a singular strategy, the quantity $h_{jj,kk}(\boldsymbol{v}^*)$ captures whether selection on trait $(j, k)$ is locally stabilising (if negative) or disruptive (if positive) when that trait evolves in isolation from all others. More generally, when $\mathbf{H}(\boldsymbol{v}^*)$ is not negative semi-definite and $\boldsymbol{v}^*$ is strongly convergence stable, selection may lead to evolutionary branching, whereby two alleles with slightly different trait values near $\boldsymbol{v}^*$ are both maintained and subsequently diverge as they accumulate further mutations (the exact conditions for this to happen when multiple traits co-evolve remains an open question, [36]).

Eqs (3a)–(7) show that under the assumption of rare mutations with small phenotypic effects, key features of the gradual long-term evolution of traits can be predicted from the selection gradients and the Jacobian and Hessian matrices. Eqs (5)–(7) assume that mutation effects on trait values are not exactly equal across traits, thereby excluding cases where genetic constraints force traits to be expressed identically across classes. We will later visit such genetically constrained trait expression, which can in fact be analysed using the same results as above, with only minimal adjustments (see appendix A.6 in S1 Text for details). In the following, we give explicit expressions for the directional and quadratic selection coefficients.

## 3 Results

### 3.1 Directional selection

#### 3.1.1 Age-specific coefficient of directional selection.
We show in appendix A.2 in S1 Text that the coefficient of directional selection on trait $(j, l)$ can be written as

$$s_{j,l}(\boldsymbol{v}) = \left[\int_0^\infty \hat{s}_{j,l}(a, \boldsymbol{v})\,\mathrm{d}a\right] \times \left.\frac{\partial z_{j,l}(u_{j,l}, v_{j,l})}{\partial u_{j,l}}\right|_{\boldsymbol{u}=\boldsymbol{v}}, \tag{8}$$

where

$$\hat{s}_{j,l}(a, \boldsymbol{v}) = \sum_{i=1}^{n_c} \nu_i^\circ \gamma_{ij} \left.\frac{\partial H_{ij}(\boldsymbol{z}_j, l_j^\circ(a), \boldsymbol{x}_j^\circ(a), \boldsymbol{v})}{\partial z_{j,l}}\right|_{\boldsymbol{z}=\boldsymbol{v}} q_j^\circ \tag{9}$$

is an age $a$ specific coefficient of directional selection. This coefficient involves three quantities. (i) The leading right-eigenvector $\boldsymbol{q}^\circ$ of the resident next-generation matrix $\mathbf{R}^\circ = \mathbf{R}(\boldsymbol{v}, \boldsymbol{v})$, such that $q_j^\circ$ gives the frequency of alleles carried by class-$j$ individuals among resident newborns. (ii) The leading left-eigenvector $\boldsymbol{\nu}^\circ$ of $\mathbf{R}^\circ$, such that $\nu_i^\circ$ is the reproductive value of an allele found in a resident newborn of class $i$ (i.e., the expected number of copies this allele will produce over the lifetime of its carrier). We refer to $\boldsymbol{q}^\circ$ and $\boldsymbol{\nu}^\circ$ as allelic class frequencies and reproductive values at birth, respectively. Finally, (iii) the marginal effect $\partial H_{ij}/\partial z_{j,l}$ of variation in trait expression on the so-called Hamiltonian $H_{ij}$, which we detail below in eq (10). For now simply note that this derivative measures how a change in trait $(j, l)$ affects the production of class-$i$ offspring by a class-$j$ individual: directly, through a change in fecundity at age $a$; and indirectly through changes in survival and development at that same age (i.e., influencing future reproduction).

With these definitions in mind, $\hat{s}_{j,l}(a, \boldsymbol{v})$ in eq (9) can be read from right to left as follows. First, $q_j^\circ$ gives the probability that a mutant allele occurs in a newborn of class $j$, so that a variation in trait $(j, l)$ will be expressed in such an individual and affect its life history across all ages $a$. Second, $\partial H_{ij}/\partial z_{j,l}$ quantifies how such age-specific variation affects the production of class-$i$ offspring, accounting for both current and future contributions, i.e., at age $a$ and later. Third, $\gamma_{ij}$ converts offspring into transmitted mutant allele copies, and finally $\nu_i^\circ$ weights each such descendant copy by its reproductive value in class $i$. Summing over all offspring classes $i$ gives the total fitness consequences of variation in trait $(j, l)$ via its effects on fecundity, survival and development at age $a$. Directional selection on trait $(j, l)$ is finally obtained by integrating these age-specific coefficients over all ages (eq 8).

### 3.1.2 The Hamiltonian: partitioning age-specific fitness effects.
Quantifying the fitness effects at a specific age relies on the Hamiltonian:

$$H_{ij}(\boldsymbol{z}_j, l_j(a), \boldsymbol{x}_j(a), \boldsymbol{v}) = l_j(a)f_{ij}(\boldsymbol{z}_j, \boldsymbol{x}_j(a), \boldsymbol{v}) - \lambda_{ij}^{l}(a)\mu_j(\boldsymbol{z}_j, \boldsymbol{x}_j(a), \boldsymbol{v})l_j(a)$$
$$+ \boldsymbol{\lambda}_{ij}^{x}(a) \cdot \boldsymbol{g}_j(\boldsymbol{z}_j, \boldsymbol{x}_j(a), \boldsymbol{v}),$$

(10)

where $\lambda_{ij}^{l}(a)$ and $\boldsymbol{\lambda}_{ij}^{x}(a) = (\lambda_{ij,1}^{x}(a), ..., \lambda_{ij,n_x}^{x}(a))$ are so-called costate variables that are given by the solution to the system of ODEs:

$$\frac{d\lambda_{ij}^{l}(a)}{da} = - \left.\frac{\partial H_{ij}(\boldsymbol{z}_j, l_j(a), \boldsymbol{x}_j^\circ(a), \boldsymbol{v})}{\partial l_j(a)}\right|_{z=v} \qquad \text{with i.c.} \qquad \lambda_{ij}^{l}(0) = R_{ij}^\circ,$$

$$\frac{d\lambda_{ij,m}^{x}(a)}{da} = - \left.\frac{\partial H_{ij,m}(\boldsymbol{z}_j, l_j^\circ(a), \boldsymbol{x}_j(a), \boldsymbol{v})}{\partial x_{j,m}(a)}\right|_{z=v} \qquad \text{with f.c.} \qquad \lim_{a \to \infty} \lambda_{ij,m}^{x}(a) = 0.$$

(11)

The costates $\lambda_{ij}^{l}(a)$ and $\lambda_{ij,m}^{x}(a)$ respectively measure how marginal changes in survival and in internal state $m$ at age $a$ affect the expected remaining reproductive output of a class-$j$ individual via offspring of class $i$. In particular $\lambda_{ij}^{l}(a)$ is a class-oriented equivalent of Fisher's definition of the reproductive value of an individual of age $a$ (see Box 1). The boundary conditions in eq (11) are obtained as follows. At birth, and provided fecundity is finite, we have $\lambda_{ij}^{l}(0) = R_{ij}^\circ$; namely, the resident expected lifetime class-oriented fecundity (e.g., in a haploid population without class structure $\lambda_{ij}^{l}(0) = 1$ owing to demographic consistency; see appendix A.2.3 in S1 Text). Biologically, this reflects that death at birth would eliminate the entire reproductive potential of the individual. By contrast, the development costates $\lambda_{ij,m}^{x}(a)$ vanish as $a \to \infty$, under the assumption that traits do not affect internal states at birth but do affect their finite terminal values (see appendix A.2.3 in S1 Text). Intuitively, this is because at very old ages the expected remaining reproductive output approaches zero, such that marginal changes in state values no longer influence fitness.

---

### Box 1. Costates and reproductive values in age- and class-structured populations.

Consider the remaining number of offspring of class $i$ that a mutant individual of class $j$ and age $a$ is expected to produce until its death,

$$\hat{R}_{ij}(\boldsymbol{z}_j, l_j(a), \boldsymbol{x}_j(a), \boldsymbol{v}, a) = \int_a^\infty l_j(\tau)f_{ij}(\boldsymbol{z}_j, \boldsymbol{x}_j(\tau), \boldsymbol{v}) \, d\tau,$$

(I-A)

such that $\hat{R}_{ij}(\boldsymbol{z}_j, l_j(0), \boldsymbol{x}_j(0), \boldsymbol{v}, 0) = R_{ij}(\boldsymbol{u}_j, \boldsymbol{v})$ (see eq 1). Using results from [9] (their appendix B.1), we obtain that the costates $\lambda_{ij}^{l}(a)$ and $\boldsymbol{\lambda}_{ij}^{x}(a)$ satisfy

$$\lambda_{ij}^l(a) = \left. \frac{\partial \hat{R}_{ij}(\mathbf{z}_j, l_j(a), \mathbf{x}_j^\circ(a), \mathbf{v}, a)}{\partial l_j(a)} \right|_{\mathbf{z}=\mathbf{v}} \quad \text{and} \quad \lambda_{ij}^x(a) = \left. \frac{\partial \hat{R}_{ij}(\mathbf{z}_j, l_j^\circ(a), \mathbf{x}_j(a), \mathbf{v}, a)}{\partial \mathbf{x}_j(a)} \right|_{\mathbf{z}=\mathbf{v}}. \tag{I-B}$$

Costates can therefore be interpreted as the change in the expected remaining class-oriented reproductive output of an individual of class $j$ and age $a$, resulting from a change in its current survivorship or internal states values. The costate associated to survivorship, in particular, can be expressed as

$$\lambda_{ij}^l(a) = \frac{1}{l_j^\circ(a)} \hat{R}_{ij}(\mathbf{v}_j, l_j^\circ(a), \mathbf{x}_j^\circ(a), \mathbf{v}, a). \tag{I-C}$$

We see from eq (I-C) that $\lambda_{ij}^l(a)$ is the expected remaining class-oriented reproductive output of a resident individual, conditional on this individual having survived to age $a$. Thus, $\lambda_{ij}^l(a)$ is the class-oriented equivalent of Fisher's reproductive value [37–39]. Note that, since according to eq (I-C) we have $\lambda_{ij}^l(0) = \hat{R}_{ij}(\mathbf{v}_j, l_j^\circ(0), \mathbf{x}_j^\circ(0), \mathbf{v}, 0) = R_{ij}(\mathbf{v}_j, \mathbf{v})$, we obtain from the definition of the left-eigenvector $\boldsymbol{\nu}^\circ$ of $\mathbf{R}^\circ$ (eq. A-3) that

$$\nu_j^\circ = \sum_{i=1}^{n_c} \nu_i^\circ \gamma_{ij} \lambda_{ij}^l(0). \tag{I-D}$$

This connects costates to the elements $\nu_j^\circ$ of the leading left-eigenvector of the next-generation matrix (i.e., allelic reproductive values at birth). While costates are age-specific measures of future reproductive output at the individual level, reproductive values in $\boldsymbol{\nu}$ are lifetime measures at the allelic level that take into account the value of each type of descendants (i.e., $\nu_i^\circ$ in eq I-D).

The Hamiltonian eq (10) quantifies how a class-$j$ individual, at age $a$, contributes to its lifetime production of class-$i$ offspring through three distinct components: (1) current offspring production through fecundity $f_{ij}$; (2) future reproductive output depending on mortality rate $\mu_j$ at that age, weighted by $\lambda_{ij}^l(a)$; and (3) future reproductive output depending on development rate $\mathbf{g}_j$ at that age, weighted by $\boldsymbol{\lambda}_{ij}^x(a)$. This decomposition reflects the fact that at any given age, an individual can influence its eventual production of class-$i$ offspring not only through its immediate fecundity, but also through its current survival and developmental rates, which determine its future reproduction. The Hamiltonian can thus be interpreted as an age-specific measure of reproductive success—one that, when integrated over age (as in eq 8), allows us to quantify the total effect of a trait change on an individual's lifetime reproductive success.

An important practical point is that in the coefficient of directional selection (eq 9) the derivative of the Hamiltonian is evaluated at resident trait values. As a result, when computing the Hamiltonian and solving the ODEs for the costates (eq 11), we only need to consider resident survivorship and state dynamics. Concretely, it is sufficient to solve

$$\begin{aligned}
\frac{dl_j^\circ(a)}{da} &= -\mu_j(\mathbf{v}_j, \mathbf{x}_j^\circ(a), \mathbf{v})\, l_j^\circ(a) \quad \text{with i.c.} \quad l_j^\circ(0) = 1, \\
\frac{d\mathbf{x}_j^\circ(a)}{da} &= \mathbf{g}_j(\mathbf{v}_j, \mathbf{x}_j^\circ(a), \mathbf{v}) \quad \text{with i.c.} \quad \mathbf{x}_j^\circ(0) = \mathbf{x}_{j,0}.
\end{aligned} \tag{12}$$

rather than the mutant system in eq (2). This can greatly simplify the analysis as we will illustrate later in our example.

### 3.1.3 Life history trade-offs in structured population.

The age-specific coefficient of directional selection (eq 9) is expressed in terms of the effect of a trait change on the Hamiltonian, which in turn separates the relevant fitness components at each age. To gain further insights, one can substitute eq (10) into eq (9), giving

$$
\hat{s}_{j,l}(a, \boldsymbol{v}) = \sum_{i=1}^{n_c} \nu_i^\circ \gamma_{ij} \left( l_j^\circ(a) \left. \frac{\partial f_{ij}(\boldsymbol{z}_j, \boldsymbol{x}_j^\circ(a), \boldsymbol{v})}{\partial z_{j,l}} \right|_{\boldsymbol{z}=\boldsymbol{v}} \right.
$$
$$
\left. - \lambda_{ij}^{\mathsf{l}}(a) l_j^\circ(a) \left. \frac{\partial \mu_j(\boldsymbol{z}_j, \boldsymbol{x}_j^\circ(a), \boldsymbol{v})}{\partial z_{j,l}} \right|_{\boldsymbol{z}=\boldsymbol{v}} + \boldsymbol{\lambda}_{ij}^{\mathsf{x}}(a) \cdot \left. \frac{\partial \boldsymbol{g}_j(\boldsymbol{z}_j, \boldsymbol{x}_j^\circ(a), \boldsymbol{v})}{\partial z_{j,l}} \right|_{\boldsymbol{z}=\boldsymbol{v}} \right) q_j^\circ .
$$
(13)

Eq (13) highlights the fundamental trade-off between current and future reproduction, while accounting for age, class structure, ontogeny and ploidy. This is made explicit by showing that directional selection boils down to a weighted effect of a trait change on the three core life-history components: fecundity, mortality, and development. The weights $q_j^\circ$, $\nu_i^\circ$ and $\gamma_{ij}$ capture the relative importance of different parent and offspring classes, $l_j^\circ(a)$ weights age-specific effects by the probability of surviving to age $a$, and the costates $\lambda_{ij}^{\mathsf{l}}(a)$ and $\boldsymbol{\lambda}_{ij}^{\mathsf{x}}(a)$ measure how a trait change at age $a$ influences future reproduction through survival and development. Taken together, these weights quantify how selection balances all the potential first-order effects of trait expression. They make it possible to distinguish three types of trade-offs: (i) within-age trade-offs, between the contributions to fecundity, survival or development at a given age (the term within parentheses in eq 13); (ii) among-class trade-offs, reflecting the relative value of producing offspring of different classes (arising from the summation over classes in eq 13); and (iii) among-age trade-offs, arising when eq (13) is integrated over ages in eq (8), and which depend on how survivorship $l_j^\circ(a)$ declines with age and on how the costates $\lambda_{ij}^{\mathsf{l}}(a)$ and $\boldsymbol{\lambda}_{ij}^{\mathsf{x}}(a)$ vary with age— either decreasing or increasing depending on developmental effects and life history.

### 3.1.4 Connections with previous expressions for directional selection.

Eq (13) connects to several results from the literature. First, in the absence of ontogeny, it reduces to the standard selection gradient in class-structured populations [40–43]. Second, in the absence of class structure, it reduces to the selection gradient on constant traits (controls) affecting life history [9,11,12], extended to to arbitrary ploidy. Third, the term within parenthesis in eq (13) is conceptually similar to expressions for the fitness effects of age-specific trait expression under complete plasticity (usually derived in the absence of class structure, e.g., [1–3,8,38,44–46], but see [47]).

Finally, when mutations act only on mortality at age $a$, eq (13) reduces to Hamilton's first indicator of the strength of selection ([13], eq. 9, p. 17; see also [48]), with selection being proportional to $l_j^\circ(a)\lambda_{ij}^{\mathsf{l}}(a) = \int_a^\infty l_j^\circ(\tau) f_{ij}(\boldsymbol{v}_j, \boldsymbol{x}_j^\circ(\tau), \boldsymbol{v}) \, \mathrm{d}\tau$, i.e., to the expected remaining reproductive output (eq. I-C); while for mutations acting only on fecundity at age $a$, eq (13) reduces to Hamilton's second indicator, i.e., survivorship to age $a$, $l_j^\circ(a)$. Eq (13) shows that selection generally acts not only through mortality and fecundity, but through development as well, taking into account the specificities of those processes within each class. A relevant implication of including development, in particular, is that it can significantly alter among-age trade-offs between life-history components. Indeed, developmental effects can be strongly non-monotonic with respect to age (e.g., during metamorphosis), and as a result the costates associated with internal states do not necessarily decline with age. Consequently, the age-specific strength of selection also need not decrease monotonically with age [49,50], contrary to a common interpretation of Hamilton's seminal results [13]. We illustrate this in our example model (Section 4).

## 3.2 Quadratic selection

Once the population expresses a singular trait value $\boldsymbol{v}^*$ where directional selection vanishes (i.e., such that eq 5 holds), the nature of selection is given by the class-oriented Hessian matrices $\mathbf{H}_{ij}(\boldsymbol{v}^*)$ (eq 3c). We show in appendix A.3 in S1 Text that the $(k, l)$-element of these matrices can be expressed as

$$h_{ij,kl}(\mathbf{v}^*) = \left[\int_0^\infty \hat{h}_{ij,kl}(a, \mathbf{v}^*)da\right] \times \left.\frac{\partial z_{i,k}(u_{i,k}, v_{i,k})}{\partial u_{i,k}}\right|_{\mathbf{u}=\mathbf{v}=\mathbf{v}^*} \times \left.\frac{\partial z_{j,l}(u_{j,l}, v_{j,l})}{\partial u_{j,l}}\right|_{\mathbf{u}=\mathbf{v}=\mathbf{v}^*},$$

(14)

where $\hat{h}_{ij,kl}(a, \mathbf{v}^*)$ is an age-specific coefficient of quadratic selection, measuring the effects of a joint change in traits $(i, k)$ and $(j, l)$ on fitness through their influence on life-history components at age $a$ (i.e., fecundity, survival, and development), that can be decomposed as

$$\hat{h}_{ij,kl}(a, \mathbf{v}^*) = \begin{cases} \hat{h}_{jj,kl}^{zz}(a, \mathbf{v}^*) + \hat{h}_{jj,kl}^{xx}(a, \mathbf{v}^*) + \hat{h}_{jj,kl}^{zx}(a, \mathbf{v}^*) \\ \quad + \hat{h}_{jj,kl}^{zl}(a, \mathbf{v}^*) + \hat{h}_{jj,kl}^{lx}(a, \mathbf{v}^*) + \hat{h}_{jj,kl}^{zq}(a, \mathbf{v}^*) & \text{if } i = j \\ \hat{h}_{jj,kl}^{zq}(a, \mathbf{v}^*) & \text{if } i \neq j. \end{cases}$$

(15)

Each term captures a relevant age-specific effect that can generate correlational selection among traits $(i, k)$ and $(j, l)$ (or disruptive selection on a trait when $i = j$ and $k = l$) that we detail below.

### 3.2.1 Age-specific direct trait effects.

The first element in the right-hand side of eq (15) with $i = j$ is given by

$$\hat{h}_{jj,kl}^{zz}(a, \mathbf{v}^*) = \sum_{m=1}^{n_c} \nu_m^\circ \gamma_{mj} \left.\frac{\partial^2 H_{mj}(\mathbf{z}_j, l_j^\circ(a), \mathbf{x}_j^\circ(a), \mathbf{v})}{\partial z_{j,k} \partial z_{j,l}}\right|_{\mathbf{z}=\mathbf{v}=\mathbf{v}^*} q_j^\circ.$$

(16a)

Eq (16a) describes the direct joint effects of traits $(j, k)$ and $(j, l)$ on the Hamiltonian $H_{mj}$ (eq 10), summed over all offspring classes $m$ with appropriate weights. Biologically, the derivative in eq (16a) quantifies whether simultaneous changes in the two traits, expressed in the same individual of class $j$, increase or decrease its production of class-$m$ offspring. As with directional selection (eq 13), these fitness effects can manifest through fecundity at age $a$ or through survival and development at that same age, which influence future reproduction.

Eq (16a) can thus be viewed as the direct extension of the classical measure of correlational selection (i.e., described in the absence of any population structure, [51]). In that setting, correlational selection is due to multiplicative effects of traits on reproduction only. By contrast, correlational selection here can also arise because traits have multiplicative effects on survival rates (i.e., on $\mu_j(\mathbf{z}_j, \mathbf{x}_j(a), \mathbf{v})$) or on development rates (i.e., on $g_j(\mathbf{z}_j, \mathbf{x}_j(a), \mathbf{v})$) at age $a$. For instance, if two traits interact to determine the growth rate – say one trait sets foraging effort, which determines resource acquisition, and the other governs how much of these resources are allocated to growth – and if larger size later increases reproduction, eq (16a) indicates that selection will favour a positive association between these two traits.

These different paths, which become apparent by substituting eq (10) into eq (16a) (though we refrain from doing so explicitly for succinctness), show that population structure and ontogeny create additional opportunities for correlational selection, and thus for the emergence of polymorphism via disruptive selection. As we will now see, there are in fact many other paths for correlational selection generated by ontogeny and class structure.

### 3.2.2 Age-specific indirect effects.

The next element in the right-hand side of eq (15), which for $i = j$ is given by

$$\hat{h}_{jj,kl}^{xx}(a, \mathbf{v}^*) = \sum_{m=1}^{n_c} \nu_m^\circ \gamma_{mj} \left[\sum_{k'=1}^{n_x} \sum_{l'=1}^{n_x} \left(\frac{\partial^2 H_{mj}(\mathbf{z}_j, l_j^\circ(a), \mathbf{x}_j(a), \mathbf{v})}{\partial x_{j,k'}(a) \partial x_{j,l'}(a)} \frac{\partial x_{j,k'}(a)}{\partial z_{j,k}} \frac{\partial x_{j,l'}(a)}{\partial z_{j,l}}\right)\right|_{\mathbf{z}=\mathbf{v}=\mathbf{v}^*}\right] q_j^\circ,$$

(16b)

describes the effect of joint change in all pairs of internal states at age $a$ ($x_{j,k'}(a)$ and $x_{j,l'}(a)$), that are induced by changes in traits $(j, k)$ and $(j, l)$. Put differently, this term captures how simultaneous changes in two traits alter state variables at age $a$ in ways that synergistically influence the different components of life history—fecundity, survival, or development—where fecundity affects reproduction at that age, while survival and development determine future reproduction (as per

eq 10). As an example where eq (16b) may be relevant, consider a scenario where one trait regulates allocation to body growth while another controls allocation to sexual maturation. These respectively determine the internal state variables 'size' and 'maturity' at a given age. When particular combinations of these internal states yield higher fitness than intermediate ones (e.g., small but early-maturing individuals, or large but late-maturing ones, as observed in Atlantic salmon, [18]), eq (16b) will be positive, indicating that selection favours an association between the two traits that induce these internal state combinations.

Correlational selection may also be due to the third element in the right-hand side eq (15), which for $i = j$ reads

$$
\hat{h}_{jj,kl}^{zx}(a, \boldsymbol{v}^*) = \sum_{m=1}^{n_c} \nu_m^\circ \gamma_{mj} \left[ \sum_{k'=1}^{n_x} \left( \frac{\partial^2 H_{mj}(\boldsymbol{z}_j, l_j^\circ(a), \boldsymbol{x}_j(a), \boldsymbol{v})}{\partial z_{j,k} \partial x_{j,k'}(a)} \frac{\partial x_{j,k'}(a)}{\partial z_{j,l}} \right) \Bigg|_{\boldsymbol{z}=\boldsymbol{v}=\boldsymbol{v}^*} \right.
$$
$$
\left. + \sum_{l'=1}^{n_x} \left( \frac{\partial^2 H_{mj}(\boldsymbol{z}_j, l_j^\circ(a), \boldsymbol{x}_j(a), \boldsymbol{v})}{\partial z_{j,l} \partial x_{j,l'}(a)} \frac{\partial x_{j,l'}(a)}{\partial z_{j,k}} \right) \Bigg|_{\boldsymbol{z}=\boldsymbol{v}=\boldsymbol{v}^*} \right] q_j^\circ,
$$

(16c)

and which describes the interaction effects between traits and internal states, i.e., it captures situations where the state-mediated influence of one trait on fitness contributions at age $a$ depends on another trait. More specifically, eq (16c) shows that correlational selection associates two traits whenever some internal state varies with the value of one trait, while the effect of that internal state on the Hamiltonian depends on the other trait.

For example, suppose the first trait determines the growth and thus morphology of some sexual traits (e.g., genital morphology) and the second determines the preference for mates with a compatible morphology. In this case, correlational selection arises because the benefit of expressing a given reproductive morphology is conditional on simultaneously expressing a preference for mates with the matching morphology. Conversely, the benefit of the preference trait depends on having the compatible morphology. Eq (16c) captures this interdependence. Similar mechanisms apply whenever behaviours or other traits have state-dependent effects, such as risk-taking behaviours that depend on body condition, or condition-dependent costs of sexual traits as in handicap models.

Another interaction with traits that can contribute to correlational selection is given by the fourth element in the right-hand side of eq (15), which for $i = j$ is

$$
\hat{h}_{jj,kl}^{zl}(a, \boldsymbol{v}^*) = \sum_{m=1}^{n_c} \nu_m^\circ \gamma_{mj} \left[ \left( \frac{\partial^2 H_{mj}(\boldsymbol{z}_j, l_j(a), \boldsymbol{x}_j^\circ(a), \boldsymbol{v})}{\partial z_{j,k} \partial l_j(a)} \frac{\partial l_j(a)}{\partial z_{j,l}} \right) \Bigg|_{\boldsymbol{z}=\boldsymbol{v}=\boldsymbol{v}^*} \right.
$$
$$
\left. + \left( \frac{\partial^2 H_{mj}(\boldsymbol{z}_j, l_j(a), \boldsymbol{x}_j^\circ(a), \boldsymbol{v})}{\partial z_{j,l} \partial l_j(a)} \frac{\partial l_j(a)}{\partial z_{j,k}} \right) \Bigg|_{\boldsymbol{z}=\boldsymbol{v}=\boldsymbol{v}^*} \right] q_j^\circ.
$$

(16d)

This term describes the interaction effects between traits and survivorship to age $a$. In biological terms, correlational selection arises because of this term whenever one trait modifies survivorship to age $a$, while another affects fecundity or mortality at that age (but not growth, since growth rates do not depend on survival, see eq 10).

As an illustration, consider a trait that improves survival to later ages, and another trait that enhances reproductive success specifically at those later ages (e.g., investment in traits expressed after maturity, such as secondary sexual displays). In this case, the benefit of the survival trait is conditional on the reproductive trait, and, conversely, the benefit of the reproductive trait depends on surviving long enough to express it. Such an interaction is captured by eq (16d), with correlational selection favouring an association between longevity and late-life reproductive investment, and disruptive selection potentially arising if intermediate strategies perform poorly compared to the extremes.

Interaction effects that are entirely mediated by life history can also generate correlational selection. This occurs when trait changes influence both internal states at a certain age and survivorship to that same age, as long as the Hamiltonian depends multiplicatively on these two components. This case is captured by the fifth element in the right-hand side of eq (15) that for $i = j$ reads

$$\hat{h}^{lx}_{jj,kl}(a, \boldsymbol{v}^*) = \sum_{m=1}^{n_c} \nu^\circ_m \gamma_{mj} \left[ \sum_{k'=1}^{n_x} \left( \frac{\partial l_j(a)}{\partial z_{j,k}} \frac{\partial x_{j,k'}(a)}{\partial z_{j,l}} + \frac{\partial l_j(a)}{\partial z_{j,l}} \frac{\partial x_{j,k'}(a)}{\partial z_{j,k}} \right) \Bigg|_{\boldsymbol{z}=\boldsymbol{v}=\boldsymbol{v}^*} \right.$$
$$\left. \times \frac{\partial^2 H_{mn}(\boldsymbol{z}_j, l_j(a), \boldsymbol{x}_j(a), \boldsymbol{v})}{\partial l_j(a) \partial x_{j,k'}(a)} \Bigg|_{\boldsymbol{z}=\boldsymbol{v}=\boldsymbol{v}^*} \right] q^\circ_j.$$

(16e)

Eq (16e) shows that correlational selection arises whenever an internal state at age $a$ affects the fecundity or mortality rates (since both rates are multiplied by survivorship in eq 10), provided traits influence both survivorship and the internal state.

For example, one trait may enhance survival through investment in immune function while another governs allocation to sexual development. The benefit of improved immunity is conditional on surviving long enough to express the sexual state, while the benefit of sexual development depends on immunity ensuring survival. Eq (16e) would therefore be positive in this case, indicating that correlational selection favours a positive association between these two traits through their joint effects on survivorship and development.

The final element in the right-hand side of eq (15) is

$$\hat{h}^{zq}_{ij,kl}(a, \boldsymbol{v}^*) = \sum_{m=1}^{n_c} \nu^\circ_m \left[ \gamma_{mi} \left( \frac{\partial H_{mi}(\boldsymbol{z}_i, l^\circ_i(a), \boldsymbol{x}^\circ_i(a), \boldsymbol{v})}{\partial z_{i,k}} \frac{\partial q_i(\boldsymbol{z}, \boldsymbol{v})}{\partial z_{j,l}} \right) \Bigg|_{\boldsymbol{z}=\boldsymbol{v}=\boldsymbol{v}^*} \right.$$
$$\left. + \gamma_{mj} \left( \frac{\partial H_{mj}(\boldsymbol{z}_j, l^\circ_j(a), \boldsymbol{x}^\circ_j(a), \boldsymbol{v})}{\partial z_{j,l}} \frac{\partial q_j(\boldsymbol{z}, \boldsymbol{v})}{\partial z_{i,k}} \right) \Bigg|_{\boldsymbol{z}=\boldsymbol{v}=\boldsymbol{v}^*} \right],$$

(16f)

which consists of the product between the effect of one trait on the Hamiltonian for a given parental class and the effect of the other trait on the probability that mutant alleles are found in newborns of that parental class (i.e., on mutant allelic frequencies at birth $\boldsymbol{q}(\boldsymbol{z}, \boldsymbol{v})$). In essence, this means that correlational selection will favour an association between two traits when one increases the likelihood of being in a certain class, while the other increases reproductive success in that class. For example, one trait may affect the development of forms that are particularly suited to reproduction as a female but not as a male, while another biases the sex ratio at birth so that a larger fraction of offspring are female. In this case, eq (16f) would be positive, indicating that correlational selection favours an association between these two traits. This is because such an association leads genotypes that produce better female forms to be more often expressed in females, while those producing better male forms are more often expressed in males.

In contrast to the previous components (eqs 16a–16e), this term $\hat{h}^{zq}_{ij,kl}$ contributes to correlational selection even when $i \neq j$ (see eq 15). The reason is that a change in a trait expressed in one class $i$ can modify the allelic class frequencies at birth in another class $q_j(\boldsymbol{z}, \boldsymbol{v})$ with $i \neq j$. For example, if sex ratio is under paternal control, then a male-expressed trait that biases the proportion of daughters directly affects the allelic frequency of the female class at birth.

Taken together, the different parts of eq (15) show that correlational selection can arise not only through direct interactions among traits (eq 16a), but also indirectly via their effects on internal states, survival, or class representation at birth (eqs 16b–16f), thereby encompassing all pathways by which trait associations influence fitness in class-structured populations with ontogeny.

### 3.2.3 Trait effects on internal states, survivorship and class structure. 
The indirect pathways through which correlational selection acts (eqs 16b–16f) ultimately depend on how traits affect internal states, survivorship, and class

structure. Here we summarise how these effects, or perturbations, can be computed (see appendix A.4 in S1 Text for details).

First, we show in appendix A.4.1 in S1 Text that the effect of a trait $l$ on internal states at age $a$ in class $j$ is obtained as

$$\left. \frac{\partial \boldsymbol{x}_j(a)}{\partial z_{j,l}} \right|_{\boldsymbol{z}=\boldsymbol{v}=\boldsymbol{v}^*} = \int_0^a \Psi_j(a,\tau) \left. \frac{\partial \boldsymbol{g}_j(\boldsymbol{z}_j, \boldsymbol{x}_j^\circ(\tau), \boldsymbol{v})}{\partial z_{j,l}} \right|_{\boldsymbol{z}=\boldsymbol{v}=\boldsymbol{v}^*} \mathrm{d}\tau, \tag{17}$$

where $\Psi_j$ is the so-called fundamental matrix collecting the cumulative effects of states on their own rates of change and satisfying

$$\frac{\mathrm{d}\Psi_j(a,\tau)}{\mathrm{d}a} = \left. \frac{\partial \boldsymbol{g}_j(\boldsymbol{z}_j, \boldsymbol{x}_j(a), \boldsymbol{v})}{\partial \boldsymbol{x}_j(a)} \right|_{\boldsymbol{z}=\boldsymbol{v}=\boldsymbol{v}^*} \Psi_j(a,\tau) \text{ with i.c. } \Psi_j(\tau,\tau) = \mathbf{I}, \tag{18}$$

where $\mathbf{I}$ is the $(n_x \times n_x)$ identity matrix. Usefully, in the special case where rates of change in states are constant with respect to time (i.e., $\frac{\mathrm{d}}{\mathrm{d}a}(\partial \boldsymbol{g}_j(\boldsymbol{z}_j, \boldsymbol{x}_j(a), \boldsymbol{v})/\partial \boldsymbol{x}_j(a)) = 0$), the fundamental matrix has an explicit exponential form that can be plugged directly into eq (17) (appendix A.4.1 in S1 Text).

The effect of trait $l$ on survivorship to age $a$ in class $j$ is given by

$$\left. \frac{\partial l_j(a)}{\partial z_{j,l}} \right|_{\boldsymbol{z}=\boldsymbol{v}=\boldsymbol{v}^*} = -l_j^\circ(a) \int_0^a \left( \frac{\partial \mu_j(\boldsymbol{z}_j, \boldsymbol{x}_j(\tau), \boldsymbol{v})}{\partial \boldsymbol{x}_j(\tau)} \frac{\partial \boldsymbol{x}_j(\tau)}{\partial z_{j,l}} + \frac{\partial \mu_j(\boldsymbol{z}_j, \boldsymbol{x}_j^\circ(\tau), \boldsymbol{v})}{\partial z_{j,l}} \right)\bigg|_{\boldsymbol{z}=\boldsymbol{v}=\boldsymbol{v}^*} \mathrm{d}\tau, \tag{19}$$

that is, as the discounted integral of all past (direct and indirect) effects of trait expression on mortality.

Finally, appendix A.4.2 in S1 Text shows that derivatives of mutant allelic class frequencies at birth with respect to mutant traits at $\boldsymbol{v}^*$ can be obtained as

$$\left. \frac{\partial \boldsymbol{q}(\boldsymbol{z}, \boldsymbol{v}^*)}{\partial z_{j,l}} \right|_{\boldsymbol{z}=\boldsymbol{v}^*} = -\left( (\mathbf{R}^{\circ\mathsf{T}} - \mathbf{I})(\mathbf{R}^\circ - \mathbf{I}) + \mathbf{1}^{(n_c \times n_c)} \right)^{-1} (\mathbf{R}^{\circ\mathsf{T}} - \mathbf{I}) \left. \frac{\partial \mathbf{R}(\boldsymbol{z}, \boldsymbol{v}^*)}{\partial z_{j,l}} \right|_{\boldsymbol{z}=\boldsymbol{v}^*} \boldsymbol{q}^\circ, \tag{20}$$

where $\mathbf{1}^{(n_c \times n_c)}$ is a matrix of ones of dimension $(n_c \times n_c)$. See appendix A.4.2 in S1 Text for the derivatives of mutant allele frequencies when $\boldsymbol{v} \neq \boldsymbol{v}^*$.

Together, the above results provide all the ingredients needed to compute the indirect pathways of correlational selection in class-structured populations with ontogeny. We will return to how to combine these ingredients more precisely in section 3.3, where we summarise our results and highlight potential computational challenges.

**3.2.4 Connections with previous expressions for quadratic selection.** Eqs (14)–(16) connect to previous results on quadratic selection in class-structured populations found in the literature. First, in the absence of ontogeny, age-, and class-structure, eq (14) reduces to the standard multivariate quantitative genetics estimates of quadratic selection, where correlational selection is given by the cross derivative of individual fitness with respect to individual trait values [51–53]. Second, in the absence of ontogeny but with class structure (so that Hamiltonians reduce to fecundity rates), eq (14) reduces to the sum of eqs (16a) and (16f), consistent with the results of [54] for a single trait (see their eq 34 without relatedness). Finally, in the absence of class-structure and for a single trait, eqs (14)–(16) reduce to eq (3.13) of [12] (which can be recovered by substituting the Hamiltonian into our expressions). The model therefore extends these results not only to class-structured populations but also to arbitrary ploidy and any number of evolving traits. Moreover, whereas [54, eq. 35] provided a system of equations for the perturbation of class frequencies in terms of resident frequencies and derivatives of fitness components (that cannot always be solved by direct methods), and whereas [12] left the effects of traits on internal states and survivorship implicit, here we resolved these expressions fully (see section 3.2.3). In particular,

eq (20) shows that the perturbation of class frequencies can be expressed directly in terms of resident frequencies and derivatives of fitness components. This then allows for a complete characterisation of directional, correlational, and disruptive selection in class-structured populations with and without ontogeny, opening the way for the analysis of adaptation under many biologically relevant scenarios.

### 3.3  Summary of the analysis of selection

Building on the above, we now outline how the different components of our model come together in practice. In particular, we summarise the steps required to evaluate directional and quadratic selection, and to classify evolutionary singular points (e.g., [17]).

Once eq (1) and (2) have been specified for a given biological scenario, the analysis proceeds as follows. First, the selection gradient is characterised using eqs (8) and (13), which requires solving for resident costate dynamics (eq 11) and state dynamics (eq 12). This allows identification of singular trait values $v^*$ (by solving eq 5) and assessment of their convergence stability (via the Jacobian matrix, eq 6). Second, quadratic selection is characterised using eqs (14)–(20), which makes it possible to determine whether convergence stable trait values are uninvadable or candidate evolutionary branching points leading to polymorphism (when the Hessian matrix is negative-definite or not negative semi-definite, respectively). While some calculations simplify when class proportions at birth are constant (i.e., trait independent, see Box 2), in most cases internal states and costate dynamics, as well as internal states and survivorship perturbations, cannot be solved analytically. This makes numerical approaches unavoidable.

To this end, we provide a set of Mathematica [55] functions that can help perform these computations numerically. These functions only require the specification of mutant developmental and vital rates in terms of traits, survivorship and internal state values (eq 2 and fecundity rates $f_{ij}(z_j, x_j(a), v)$). Given a set of parameter values, the workflow is as follows. The selection gradient is first computed on a grid of resident allelic values $v$ by constructing Hamiltonians, solving eqs (11) and (12) numerically, and evaluating eqs (8) and (13). This gradient is then interpolated numerically, its zeros identified as singular trait values $v^*$, and the Jacobian matrix approximated numerically at these $v^*$ to identify convergence stable trait values (eq 6). Finally, the Hessian matrix at $v^*$ is computed using eq (14), via a separate calculation of each term in eq (15) (from eqs 16–20). This procedure accommodates any number of traits, and can exploit computational shortcuts available when class proportions at birth are constant (Box 2).

---

**Box 2.  Constant class proportions at birth.**

In the general case described in the main text, computing selection (eq 8 or eq 14) involves solving eq (11) for each offspring class $i$ and parental class $j$ (i.e., a total system of size $n_c \times n_c$). We show in appendix A.5.1 in S1 Text that this problem can be reduced to solving only $n_c$ equations (i.e., one per parent class $j$) under the assumption that individuals of class $j$ allocate a constant proportion $c_{ij} \in [0, 1]$ of their reproductive output into individuals of class $i$, irrespective of age or state, such that their overall fecundity $f_j$ satisfies $f_{ij}(z_j, x_j(a), v) = c_{ij} f_j(z_j, x_j(a), v)$ with $\sum_{i=1}^{n_c} c_{ij} = 1$. Several biological scenarios can be reduced to the case of constant class proportion. For instance, exogenous environmental sex-determination, sex chromosome systems coupled to fair meiosis, or random dispersal at birth between habitats, all fall into this category. Then, the age-specific directional selection pressure on trait $(j, l)$ (i.e., eq 9) simplifies to

$$\hat{s}_{j,l}(a, v) = \breve{\nu}_j^\circ \left. \frac{\partial H_j(z_j, l_j^\circ(a), x_j^\circ(a), v)}{\partial z_{j,l}} \right|_{z=v} q_j^\circ,$$

(II-E)

---

where

$$\check{\nu}_j^\circ = \sum_{i=1}^{n_c} \nu_i^\circ \gamma_{ij} c_{ij}$$

(II-F)

is the allelic reproductive value at birth of one average offspring of individuals of class $j$, and where

$$H_j(\mathbf{z}_j, l_j(a), \mathbf{x}_j(a), \mathbf{v}) = l_j(a)f_j(\mathbf{z}_j, \mathbf{x}_j(a), \mathbf{v}) - \lambda_j^l(a)\mu_j(\mathbf{z}_j, \mathbf{x}_j(a), \mathbf{v})l_j(a)$$
$$+ \lambda_j^x(a) \cdot \mathbf{g}_j(\mathbf{z}_j, \mathbf{x}_j(a), \mathbf{v})$$

(II-G)

is the Hamiltonian specific to class $j$. In equation (II-G), the class-specific costates are solutions to the systems of ODEs,

$$\frac{d\lambda_j^l(a)}{da} = -\left.\frac{\partial H_j(\mathbf{z}_j, l_j(a), \mathbf{x}_j^\circ(a), \mathbf{v})}{\partial l_j(a)}\right|_{\mathbf{z}=\mathbf{v}} \quad \text{with i.c.} \quad \lambda_j^l(0) = \sum_{i=1}^{n_c} R_{ij}^\circ,$$

$$\frac{d\lambda_j^x(a)}{da} = -\left.\frac{\partial H_j(\mathbf{z}_j, l_j^\circ(a), \mathbf{x}_j(a), \mathbf{v})}{\partial \mathbf{x}_j(a)}\right|_{\mathbf{z}=\mathbf{v}} \quad \text{with f.c.} \quad \lim_{a\to\infty} \lambda_j^x(a) = \mathbf{0}.$$

(II-H)

Intuitively, these results mean that under constant class proportions at birth, the optimization of fitness reduces to the optimization of overall lifetime reproductive output.

Two caveats are worth keeping in mind. First, our analysis is based on the reproductive number $R_0$ of a mutant allele, which is sign equivalent around one to the geometric growth rate of the mutant allele, but not strictly equal to it. This means that the coefficients of selection we derive are comparable relative to one another, but in absolute terms they should be scaled by the inverse of generation time (see for instance [30]). Therefore, while our analysis is well suited to describe how natural selection acts on life-history trade-offs (e.g., allocation to growth vs. fecundity at different ages), caution is needed if one's desire is to compare selection to other evolutionary processes such as mutation or drift. This is especially important for drift, since effective population size is also influenced by generation time [39].

Second, our analysis has so far assumed that mutation effects on trait values are not exactly equal across traits thereby excluding cases where strong genetic constraints force traits to evolve identically across classes. Such fully correlated traits are expected to play a role especially in systems where class-structure recently evolved and where genetic conflicts have not been resolved. With some adjustments, it turns out that the above results can be applied to such cases. In particular, we can readily consider the case where two traits, say $(i, k)$ and $(j, l)$, are underlain by the same pleiotropic genes such that the traits always mutate in the same way (i.e., with $\eta_{i,k} = \eta_{j,l}$). In this case, it is as though we have a single trait, such that $u_{i,k} = u_{j,l}$ and $v_{i,k} = v_{j,l}$, and a singular trait value $\mathbf{v}^*$ must then satisfy

$$s_{i,k}(\mathbf{v}^*) + s_{j,l}(\mathbf{v}^*) = 0 \quad \text{while} \quad s_{m,n}(\mathbf{v}^*) = 0 \text{ for all } (m, n) \neq (i, k), (j, l)$$

(21)

under the constraint that $v_{i,k}^* = v_{j,l}^*$. Eq (21) shows how genetic constraints can impose additional trade-offs across classes. For instance, plugging eqs (8)–(9) into eq (21) reveals that when there is antagonism between classes (i.e., when $s_{i,k}(\mathbf{v}) \neq s_{j,l}(\mathbf{v})$ with $i \neq j$), a singular trait value will reflect a compromise between selection in both classes, giving more

weight to the most common class (i.e., the class $j$ with the largest $q_j^\circ$). We detail how such genetic constraints and resulting antagonisms modify the analyses of convergence stability and uninvadability in appendix A.6 in S1 Text.

## 4 Example: ontogenetic sexual niche partitioning

We now use the above results to study the joint evolution of female and male traits that govern the ontogeny of sexually relevant morphology. Our goal is twofold: (i) to illustrate the general approach and numerical procedure on a concrete biological scenario; and (ii) to gain theoretical insight into how sex-specific developmental schedules, by shaping size or structures used in competition and mate acquisition, can feed back on the strength and direction of sexual selection.

### 4.1 The model

**4.1.1 Population and traits.** We consider a diploid, dioecious, age-structured population with two classes, females and males, denoted by subscripts f and m respectively (for readability, we use these instead of using 1 and 2 to denote classes here). Sex is determined at birth with a fixed primary sex ratio $c$ (the proportion of males in newborns, see Table 3 for symbols). Class proportions at birth are thus constants in this model. This leads to several simplifications that are detailed in Box 2. In the notation introduced there, we have $c_{mf} = c_{mm} = c$ and $c_{ff} = c_{fm} = 1 - c$. Age-specific vital rates (survival, development, and offspring production) are allowed to differ between the sexes and to depend on sex-specific traits as we detail below.

We study the co-evolution of female and male traits $z_f$ and $z_m$ that regulate the allocation of resources into the growth of a sex-specific morphological, physiological, or behavioural module, represented by the internal states $x_f(a)$ and $x_m(a)$ (and as a baseline we present the case without sex or development, where $z_f = z_m = x_f(a) = x_m(a)$ in appendix B.2 in S1 Text). Biologically, the most straightforward way to think about this module is as overall body size or body mass, where female fecundity scales with size and male compatibility with females is mediated by size matching. For ease of presentation, this is the interpretation we will adopt throughout. However, the model also applies to specialised reproductive modules such as genital morphology, gamete recognition characters (e.g., fertilisation proteins), or sexually selected phenotypes (e.g., eyestalks, ritualised dances) when female choice correlates with her internal states.

**4.1.2 Growth and survival.** We assume that the rate of change in the internal state $x_j(a)$ of an individual of sex $j \in \{f, m\}$ at age $a$ does not depend on its environment (i.e., no density dependence, no environmental effect in resource availability), and write

$$\frac{\mathrm{d}x_j(a)}{\mathrm{d}a} = g_j(z_j, x_j(a)) = z_j - \alpha x_j(a) \quad \text{with i.c.} \quad x_j(0) = 0, \tag{22}$$

where $\alpha$ is the rate at which size (or mass) is lost due to physiological constraints (e.g., respiration), and $z_j = z_j(u_j, v_j) = (u_j + v_j)/2$ is the trait value of a mutant individual, corresponding to its sex-specific basal growth rate, which can be thought of as the baseline energetic investment into growth. Since there is a single trait per sex, we simplify notation by writing $z_j = z_{j,1}$, $u_j = u_{j,1}$, and $v_j = v_{j,1}$.

There is a trade-off between allocation to growth and survival such that the sex-specific mortality rate of a mutant individual of sex $j$ is

$$\mu_j(z_j) = \mu_{e,j} + \beta_j z_j^2, \tag{23}$$

where $\mu_{e,j}$ is a constant extrinsic mortality rate (e.g., due to sex-specific risks) and $\beta_j$ modulates the cost of growth on survival. The rate of change of survivorship in a focal mutant of sex $j$ and age $a$ is thus

$$\frac{\mathrm{d}l_j(a)}{\mathrm{d}a} = -\mu_j(z_j)\, l_j(a), \tag{24}$$

**Table 3. List of symbols used in section 4.**

| Symbol | Meaning |
|---|---|
| **Parameters** | |
| $c$ | Primary sex-ratio, i.e., proportion of males in offspring at birth |
| $\alpha$ | Rate at which mass is lost |
| $\mu_{e,j}$ | Extrinsic mortality rate in sex $j \in \{f, m\}$ |
| $\beta_j$ | Survival cost to growth in sex $j$ |
| $\kappa$ | Strength of sexual selection (based on size matching with females) |
| $K$ | Density-dependence parameter (modulating female fecundity) |
| **Variables/functions** | |
| $z_m(u_m, v_m)$, $z_f(u_f, v_f)$, $z_l(u_l, v_l)$ | Male, female and shared (i.e., under genetic constraints) basal growth rates |
| $x_j(a)$, $x_j^\circ(a)$ | Size at age $a$ in a mutant and resident of sex $j$ |
| $N_T^\circ$ | Resident population density |
| $\chi_j^\circ(x_j)$ | Probability density of size $x_j$ within residents of sex $j$ |
| $\tilde{C}(x_m(a), x_f)$, $C(x_m(a), x_f)$ | Absolute and relative competitive ability of a mutant male of size $x_m(a)$ in fertilising the eggs of a female of size $x_f$ |

(from eq 2).

Solving eqs (22) and (24) at resident allelic values $\boldsymbol{v}$, the size and survivorship of a resident individual of sex $j$ and age $a$ are

$$x_j^\circ(a) = \frac{1 - e^{-\alpha a}}{\alpha} v_j \quad \text{and} \quad l_j^\circ(a) = e^{-\mu_j^\circ a},$$

(25)

where $v_j = z_j(v_j, v_j)$ and $\mu_j^\circ = \mu_j(v_j)$. We show in appendix B.1 in S1 Text that the corresponding equilibrium probability density of individuals of size $x_j$ and sex $j$ in the resident population is

$$\chi_j^\circ(x_j) = \frac{\mu_j^\circ}{v_j} \left( \frac{v_j}{v_j - \alpha x_j} \right)^{\frac{\alpha - \mu_j^\circ}{\alpha}},$$

(26)

which is defined for values of $x_j$ between zero and the maximal resident size $\tilde{x}_j^\circ = v_j/\alpha$ (obtained from eq 25 in the limit of $\alpha \to \infty$ and entailing that $\int_0^{\tilde{x}_j^\circ} \chi_j^\circ(x_j) dx_j = 1$). Accordingly, the shape of the size density distribution in each sex qualitatively depends on the balance between $\alpha$ and the mortality rate $\mu_j^\circ = \mu_{e,j} + \beta_j v_j^2$. When $\alpha > \mu_j^\circ$, the density $\chi_j^\circ(x_j)$ increases with size $x_j$, indicating that many individuals survive to reach large sizes. When $\alpha < \mu_j^\circ$, the density decreases with size, indicating that most individuals die before reaching their maximum size. This is illustrated in Fig 1, and plays an important role in evolutionary dynamics.

**4.1.3 Female and male fecundity.** The remaining model components we need to specify are the female and male fecundity rates, denoted $f_f(x_f(a), \boldsymbol{v})$ and $f_m(x_m(a), \boldsymbol{v})$. We assume that a female produces eggs at a rate $f_f(x_f(a), \boldsymbol{v})$ and that each egg is fertilised (i.e., no mate or sperm limitation). Male fecundity $f_m(x_m(a), \boldsymbol{v})$ then reflects the outcome of competition among males for the fertilisation of the eggs.

For females, we assume that fecundity increases linearly with size $x_f(a)$ and decreases with total population density, for example due to competition for resources. Specifically, we assume:

PLOS Computational
Biology

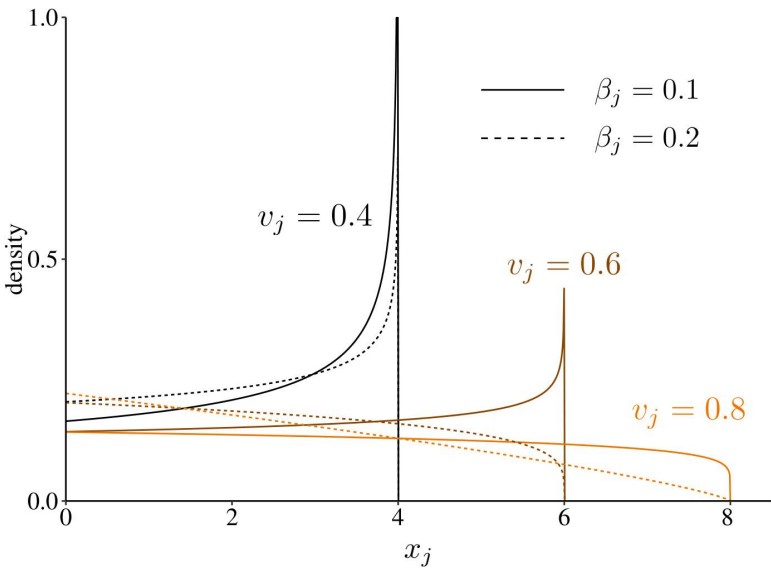

**Fig 1. Equilibrium distribution of size in the resident population.** Each line represents the equilibrium probability density of individuals of size $x_j$ and sex $j$ among residents. With increasing investment into growth, the maximum size increases (compare black, brown and orange lines), but so does the resident mortality rate $\mu_j^\circ = \mu_{e,j} + \beta_j v_j^2$, such that the proportion of smaller-sized individuals also increases. Under a stronger cost $\beta_j$ of growth on survival, the switch from a distribution with a majority of larger individuals to a population with mostly smaller individuals occurs at smaller values of $v_j^*$ (compare plain and dotted lines). Fixed parameter values: $\alpha = 0.1$ and $\mu_{e,j} = 0.05$.

$$f_f(x_f(a), \boldsymbol{v}) = x_f(a)\left(1 - \frac{N_T^\circ}{K}\right),$$

(27)

where $K > 0$ controls (inversely) the strength of density dependence (large values of $K$ mean low density-dependent competition), and $N_T^\circ$ is the equilibrium total density of individuals in the resident population (sum of male and female densities). We show in appendix B.1 in S1 Text that this is given by

$$N_T^\circ = K\left(1 - \frac{1}{(1 - c)\int_0^\infty l_f^\circ(a)x_f^\circ(a)\mathrm{d}a}\right),$$

(28)

where $(1 - c)\int_0^\infty l_f^\circ(a)x_f^\circ(a)\mathrm{d}a$ is the expected lifetime production of female offspring by a single resident female in the absence of density-dependence.

For males, we assume that fecundity — that is, the fertilisation of eggs — depends on matching with female size (e.g., due to mechanical constraints during mating or female preferences for size-similar partners) and on competition with other males. Specifically, we write the fecundity rate of a focal mutant male of size $x_m(a)$, i.e., the total number of offspring he is expected to sire across all females in the population, as

$$f_m(x_m(a), \boldsymbol{v}) = \int_0^{\tilde{x}_f^\circ} (1 - \tilde{c}^\circ)N_T^\circ \chi_f^\circ(x_f)f_f(x_f, \boldsymbol{v})C(x_m(a), x_f)\mathrm{d}x_f,$$

(29)

where we recall that $\tilde{x}_j^\circ = v_j/\alpha$ is the maximum size attainable by residents of sex $j$, and where

$$\tilde{c}^\circ = \frac{c\mu_f^\circ}{(1 - c)\mu_m^\circ + c\mu_f^\circ}$$

(30)

is the resident secondary sex-ratio (i.e., the proportion of males in the resident population; see appendix B.1.1 in S1 Text for derivation). Accordingly, $(1 - \tilde{c}^{\circ})N_T^{\circ}\chi_f^{\circ}(x_f)$ in eq (29) is the total number of females of size $x_f$, and $(1 - \tilde{c}^{\circ})N_T^{\circ}\chi_f^{\circ}(x_f)f_f(x_f, \mathbf{v})$ is the rate at which eggs are produced by these females collectively (i.e., the total fecundity rate of females of size $x_f$). For each of these eggs, we assume that the probability that it is fertilised by a focal mutant male of size $x_m(a)$ is given by

$$C(x_m(a), x_f) = \frac{\tilde{C}(x_m(a), x_f)}{\int_0^{\tilde{x}_m^{\circ}} \tilde{c}^{\circ}N_T^{\circ}\chi_m^{\circ}(x_m)\tilde{C}(x_m, x_f)\mathrm{d}x_m},$$

(31)

where the numerator gives the competitive ability of the focal male to fertilise a female of size $x_f$ and the denominator normalises over all resident males. We assume that competitive ability declines with the difference between male and female size according to

$$\tilde{C}(x_m(a), x_f) = \exp\left[-\kappa(x_m(a) - x_f)^2\right],$$

(32)

where $\kappa \geq 0$ is a parameter controlling the strength of size matching: when $\kappa = 0$, all males are equally competitive regardless of their size (i.e., $\tilde{C} = 1$); as $\kappa$ increases, fertilisation becomes increasingly size-assortative.

Equation (32) is equivalent to the compatibility function $\psi$ used in earlier models of sexual selection (e.g., [56,57]), where trait matching determines mating success and can favour polymorphism. However, whereas other models assume fixed trait values, here we apply this mating function to phenotypes that change with development over individuals' lifetimes. From a mechanistic point of view, eq (32) applies indiscriminately to systems where female directly express preference for similar-sized males (e.g., as in the convict cichlid *Archocentrus nigrofasciatus*; [58]), where they co-occur more often with similar-sized males (e.g., as in the striped goby cichlid *Eretmodus cyanostictus*; [59]), or where mating success depends on size-similarity due to morphological constraints (e.g., as in the Jerusalem cricket *Stenopelmatus sp.*; [60]). Finally, note that eq (29) links male and female fecundity such that it satisfies Fisher's condition (i.e., any newborn has one parent of each sex; [61, 62]; see appendix A.5.2 in S1 Text).

### 4.2 Analysis

**4.2.1 Hamiltonians and costates.** Since class proportions at birth are constant in this model, we only need class–specific Hamiltonians and costates (see Box 2). Substituting the model ingredients (eqs 22, 23, 27, and 29) into eq. (II-G) of Box 2 gives

$$H_f(z_f, l_f(a), x_f(a), \mathbf{v}) = l_f(a)f_f(x_f(a), \mathbf{v}) - \lambda_f^l(a)\left[\mu_{e,f} + \beta_f z_f^2\right]l_f(a) + \lambda_f^x(a)\left[z_f - \alpha x_f(a)\right]$$

$$H_m(z_m, l_m(a), x_m(a), \mathbf{v}) = l_m(a)f_m(x_m(a), \mathbf{v}) - \lambda_m^l(a)\left[\mu_{e,m} + \beta_m z_m^2\right]l_m(a) + \lambda_m^x(a)\left[z_m - \alpha x_m(a)\right],$$

(33)

where the sex-specific costates satisfy the system of ODEs:

$$\frac{\mathrm{d}\lambda_f^x(a)}{\mathrm{d}a} = \alpha\lambda_f^x(a) - l_f^{\circ}(a)\left(1 - \frac{N_T^{\circ}}{K}\right),$$

(34a)

$$\frac{\mathrm{d}\lambda_m^x(a)}{\mathrm{d}a} = \alpha\lambda_m^x(a) - l_m^{\circ}(a)\left.\frac{\partial f_m(x_m(a), \mathbf{v})}{\partial x_m(a)}\right|_{z=\mathbf{v}},$$

(34b)

$$\frac{d\lambda_f^l(a)}{da} = \mu_f^\circ \lambda_f^l(a) - x_f^\circ(a)\left(1 - \frac{N_T^\circ}{K}\right),$$

(34c)

$$\frac{d\lambda_m^l(a)}{da} = \mu_m^\circ \lambda_m^l(a) - f_m(x_m^\circ(a), \boldsymbol{v}).$$

(34d)

These have boundary conditions $\lim_{a\to\infty} \lambda_j^x(a) = 0$ and $\lambda_j^l(0) = 2$ since in a resident sexual population at demographic equilibrium, each individual has an expected lifetime reproductive output of two offspring (one daughter and one son on average). We solve these ODEs numerically.

### 4.2.2 Directional selection: balancing future effects of growth across ages.
Age-specific coefficients of directional selection are obtained by substituting eq (33) into eq. (II-E) of Box 2. In addition, since we are considering a diploid population with constant primary sex ratio, the class frequencies and reproductive values are $q_f^\circ = (1 - c)$, $q_m^\circ = c$ and $\check{\nu}_f^\circ = \check{\nu}_m^\circ = \frac{1}{2}$ respectively (see Box 2 and appendix A.5.2 in S1 Text for derivation). Using these quantities, we have

$$\hat{s}_f(a, \boldsymbol{v}) = \check{\nu}_f^\circ \left.\frac{\partial H_f(z_f, l_f^\circ(a), x_f^\circ(a), \boldsymbol{v})}{\partial z_f}\right|_{z=v} q_f^\circ = \frac{1-c}{2}\left(\lambda_f^x(a) - \lambda_f^l(a)l_f^\circ(a)2\beta_f v_f\right),$$

(35a)

$$\hat{s}_m(a, \boldsymbol{v}) = \check{\nu}_m^\circ \left.\frac{\partial H_m(z_m, l_m^\circ(a), x_m^\circ(a), \boldsymbol{v})}{\partial z_m}\right|_{z=v} q_m^\circ = \frac{c}{2}\left(\lambda_m^x(a) - \lambda_m^l(a)l_m^\circ(a)2\beta_m v_m\right).$$

(35b)

Eq (35) shows that age-specific selection at age $a$ on the basal growth rate balances two antagonistic effects on future reproduction. On the one hand, future fitness effects mediated by growth tend to favour larger $z_j$ especially at a young age (captured by $\lambda_j^x(a)$ in eq (35), see Fig 2A). On the other hand, future fitness effects mediated by survival favour smaller $z_j$ (captured by the term $\lambda_j^l(a)l_j^\circ(a)2\beta_j v_j$ in eq 35). The reason is that higher growth raises mortality and hence lowers the expected remaining reproductive output, $\lambda_j^l(a)l_j^\circ(a)$ here, which itself declines with age (Fig 2B). In our model, $\lambda_j^x(a)$ declines more steeply than $\lambda_j^l(a)l_j^\circ(a)$ with age $a$, so the balance between these two antagonistic effects changes with age: selection favours faster growth at younger ages but slower growth later in life (Fig 2C).

Selection eventually favours a compromise between these antagonistic effects. This is formalised by unique convergence stable trait values in females and males $\boldsymbol{v}^* = (v_f^*, v_m^*)$ that satisfy

$$v_j^* = \frac{1}{2\beta_j}\frac{\int_0^\infty \lambda_j^x(a)da}{\int_0^\infty \lambda_j^l(a)l_j^\circ(a)da}$$

(36)

for sex $j$ (obtained by plugging eq 35 into eq 8). Note that eq (36) is implicit as both costates and internal states depend on $\boldsymbol{v}^*$. It nevertheless makes clear that the equilibrium growth rate increases when the cumulative benefits of growth (i.e., $\int_0^\infty \lambda_j^x(a)da$) outweigh the cumulative benefits of survival (i.e., $\int_0^\infty \lambda_j^l(a)l_j^\circ(a)da$).

A numerical analysis of eq (36) together with eq (34) allows us to investigate how key parameters influence $\boldsymbol{v}^*$ and thus life histories and demography at evolutionary equilibrium. Unsurprisingly, greater growth costs $\beta_j$ select for slower growth in both females (Fig 3A) and males (S1A Fig), leading to longer lives with delayed reproduction. By contrast, higher extrinsic mortality $\mu_{e,j}$ reduces the expected remaining reproductive output $\lambda_j^l(a)l_j^\circ(a)$ and therefore selects for faster growth in sex $j$ (Fig 3B for females, S1B Fig for males), leading to "live-fast–die-young" types of life histories [63]. In females, faster growth partly compensates for the reduction in average size (and hence fecundity) caused by extrinsic mortality, such that population extinction ($N_T^\circ \leq 0$) occurs only at relatively high values of $\mu_{e,f}$ (Fig 3B, dark gray area).

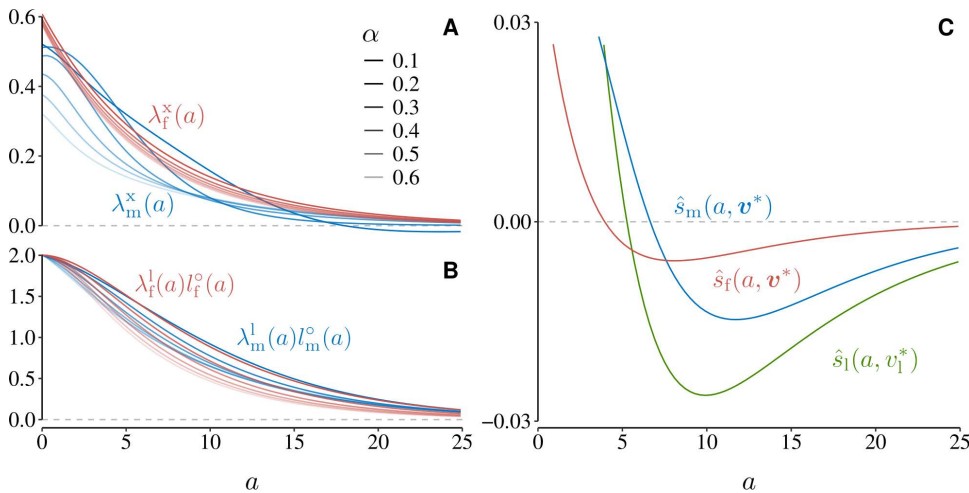

**Fig 2. Age-specific directional selection arises from opposing effects of growth on reproductive value and survival. A.** Age-specific benefit of increased size, captured by the costate $\lambda_f^x(a)$, declines with age. This reflects the diminishing marginal returns of size on future reproduction as individuals age, especially when size loss is high ($\alpha$ large). **B.** Expected remaining reproductive output at age $a$, given by $\lambda_j^l(a)l_j^\circ(a)$, also declines with age. However, it is more sensitive to $\alpha$ in females than in males (lighter curves). This is because higher $\alpha$ reduces the maximum attainable size and thus lifetime fecundity in females. These opposing age-specific effects (A vs. B) jointly generate the pattern of directional selection in panel C: early selection favours growth due to future size benefits, while late selection disfavours it due to survival costs. **C.** Age-specific directional selection on growth (from eqs 35 and 38) at the convergence stable strategies ($\boldsymbol{v}^* = (v_f^*, v_m^*) = (0.518, 0.432)$ for independent traits; $v_l^* = 0.356$ for a shared trait). Each curve integrates to zero (owing to eqs 5 and 8), but its shape reveals when selection favours increased or decreased allocation to growth. Selection tends to favour faster growth at early ages but reduced allocation later in life. Parameter values: $c = 0.5$, $\mu_{e,f} = \mu_{e,m} = 0.1$, $\beta_f = \beta_m = 0.2$, $\kappa = 1$, and $K = 10^4$. Parameter $\alpha$ has value 0.2 in panel **C**.

The parameter $\alpha$, which determines the rate at which size is lost (eq 22), has opposite consequences in the two sexes. In females, greater $\alpha$ selects for faster growth (larger $v_f^*$), while in males it selects for slower growth (smaller $v_m^*$; Fig 3C). The reason is that when $\alpha$ is large, it prevents males and females from reaching higher sizes. This is reflected in the size distributions being shifted towards smaller values when $\alpha$ is increased (compare the histograms of Fig 4A and 4C). In females, this prevents individuals from reaching high fecundity, lowering their remaining reproductive output at any given age (Fig 2B), and thus favouring growth over survival. For males, meanwhile, because greater $\alpha$ reduces average female size and male success depends on matching female size (eq 32), this favours slower growth as long as $\kappa > 0$.

In fact, the strength of size-based competition $\kappa$ plays a decisive role in male evolution. When $\kappa = 0$, size brings no reproductive benefit and males are selected to invest nothing in growth ($v_m^* = 0$). When $\kappa$ is large, size matching is crucial for male fitness and selection favours growth rates that allow males to match female size. Simulations confirm this: $\kappa$ has no effect on female evolution $v_f^*$ (red curves, left vs. right columns in Fig 4), but $v_m^*$ evolves to be larger when $\kappa$ is large (left curves, left vs. right columns in Fig 4), thereby reducing the mismatch between female and male size distributions (compare red and blue histograms in Fig 4).

**4.2.3 Quadratic selection: sexual selection based on trait matching favours life-history polymorphism in males.** Once the population has converged to the equilibrium $\boldsymbol{v}^*$ characterized by eq (36), does it remain monomorphic under stabilising selection or can it become polymorphic due to disruptive selection? Numerical evaluations of eq (14) show that the Hessian matrix can be positive-definite, and thus selection can be disruptive, but only when

$$\hat{h}_{mm}^{xx}(a, \boldsymbol{v}^*) = \frac{\kappa(1 - e^{-\alpha a})^2}{2(1-c)\alpha^2}l_m^\circ(a)$$

$$\times \left[\int_0^{\tilde{x}_f^\circ}(1 - \tilde{c}^\circ)N_T^\circ \chi_f^\circ(x_f)f_f(x_f, \boldsymbol{v}^*)C(x_m^\circ(a), x_f)\left(2\kappa(x_m^\circ(a) - x_f)^2 - 1\right)dx_f\right]$$

$$(37)$$

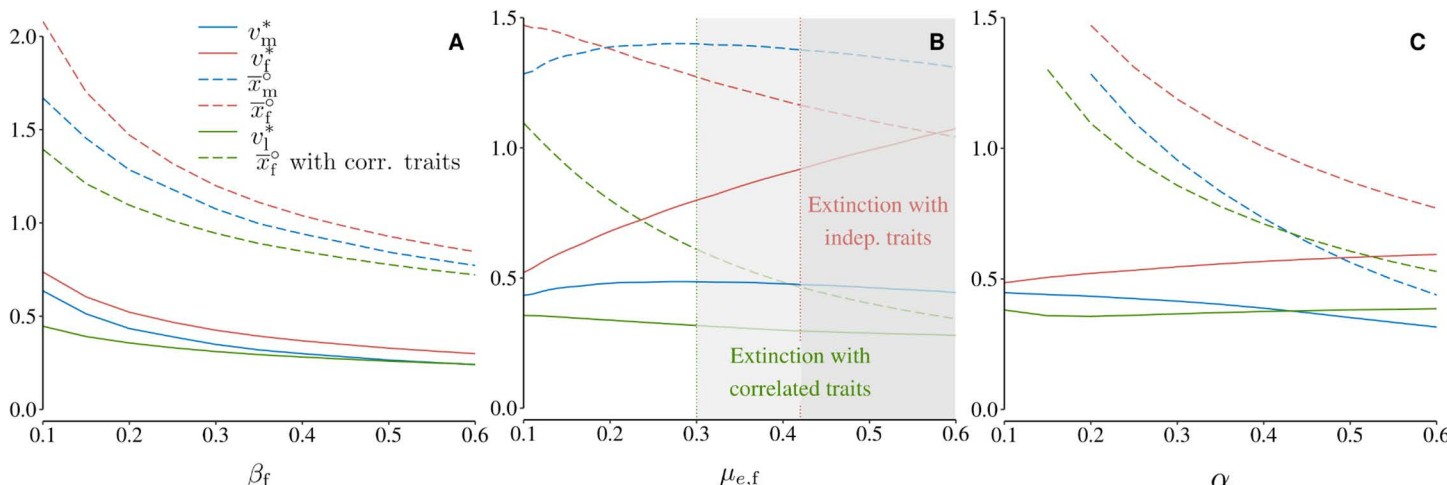

**Fig 3. Female mortality and growth parameters influence sex-specific evolution and extinction risk.** Each panel shows convergence stable growth rates $v^*$ (solid lines) and corresponding average sizes (dashed lines) for females (red) and males (blue), as a function of the parameter indicated on the x-axis (computed from eq 36 for trait values and B-99 for average sizes). Trait values under genetic constraint (i.e., identical growth rate in males and females, $v_i^*$) and the resulting female size are shown in green (from eq 39). **A.** Higher cost of growth to survival in females ($\beta_f$) selects for slower growth and smaller average size. Males evolve smaller size in response, to match females owing to size-matching sexual selection. **B.** Higher female extrinsic mortality ($\mu_{e,f}$) favours faster growth. Above a threshold, female fecundity becomes insufficient to maintain recruitment and the population goes extinct ($N_T^\circ \leq 0$; dark grey area). This extinction threshold is lower when genetic correlations constrain male and female growth to evolve identically (light grey area), because sexual antagonism prevents females from evolving growth rates high enough to offset increased mortality (green). Note that convergence stable growth rates $v^*$ and average sizes are reported beyond the extinction thresholds (transparent lines) for graphical purposes, since the model provides result even when $N_T^\circ \leq 0$. **C.** A higher rate of size loss ($\alpha$) prevents individuals from reaching larger sizes. This favours faster growth in females, but slower growth in males. See main text for explanation. Parameter values (unless varied on x-axis): $c = 0.5$, $\alpha = 0.2$, $\mu_{e,f} = \mu_{e,m} = 0.1$, $\beta_f = \beta_m = 0.2$, $\kappa = 1$, and $K = 10^4$.

is positive and large, i.e., when positive multiplicative effects of size on male fitness at age $a$ induced by changes in growth rate are large (eq 16b, as all other components of eq 15 were found to be too small to drive disruptive selection). Such multiplicative effects via size suggest that disruptive selection should favour males with either fast or slow growth such that small and large males of the same age coexist. Individual-based simulations confirm this (appendix B.3 in S1 Text for procedure). These show evolutionary branching leading to the coexistence of two diverged alleles coding for different growth rates, such that three types of males occur: two homozygotes with opposite growth strategies and an intermediate heterozygote (Fig 4D, blue dots). Some males thus grow substantially slower, remaining smaller for most of their lives but being longer-lived. At the population level this produces a multimodal male size distribution and a closer match with the female size distribution (compare panels C and D of Fig 4).

A closer inspection of eq (37) reveals the drivers of polymorphism. Disruptive selection on male growth is favoured when the number of competitors is large (i.e., when $c$ is high or male survivorship is high), when size matching is strong (large $\kappa$), and when individuals can attain large sizes (small $\alpha$). In addition, the integral in eq (37) shows that a necessary condition is that the fecundity-weighted average of $(2\kappa(x_m^\circ(a) - x_f)^2 - 1)$ is positive, which occurs when male and female size distributions diverge. In particular, large positive values of $\hat{h}_{mm}^{xx}(a, v^*)$, and thus disruptive selection, occur when males are predominantly large ($\alpha > \mu_m^\circ$; see Fig 1) whereas females are predominantly small ($\alpha < \mu_f^\circ$). In this case, at the convergence stable trait value $v_m^*$ most males are close to the average female size, while only a few match the smaller sizes more common in females (see Fig 4C). Under high $\kappa$, this leads to intense competition among larger males and thus to a fitness advantage for rare smaller males. Polymorphism therefore evolves because males specialize: some grow rapidly and die young, specializing into fertilising larger and more fecund females, while others grow slowly and live longer, and

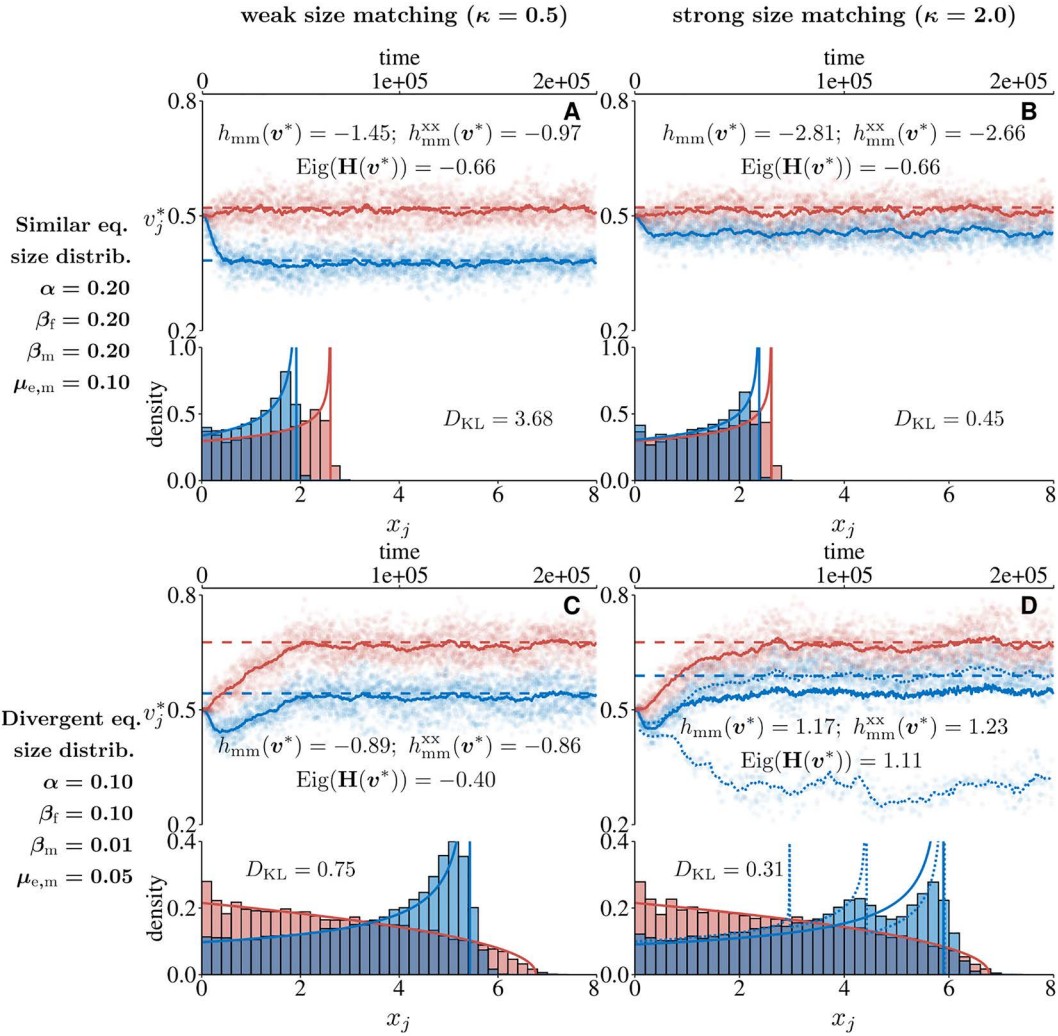

**Fig 4. Evolutionary branching of male growth generates male life-history polymorphism.** Each panel shows results from individual-based simulations (appendix B.3 in S1 Text). The top part shows the expressed trait values in females (red) and males (blue): randomly sampled allelic values (dots; 10 copies every 10 time units), population mean (solid lines), and analytically predicted convergence stable trait values (dashed lines). Reported values give: the male-specific component $h_{mm}(\boldsymbol{v}^*)$ (eq 14); its size-mediated component $h_{mm}^{xx}(\boldsymbol{v}^*) = 0.25 \int_0^\infty \hat{h}_{mm}^{xx}(a, \boldsymbol{v}^*)\,da$ (see eqs 37 and A-37, with 0.25 for diploidy); and the leading eigenvalue of $\mathbf{H}(\boldsymbol{v}^*)$. As expected, when this eigenvalue is negative, the population remains monomorphic (panels A–C). When it is positive, evolutionary branching occurs in males (panel D, split in blue lines). The bottom part of each panel shows male and female size distributions at equilibrium as histograms (averaged between time steps $10^5$ and $2 \times 10^5$), along with the analytically predicted distribution at $\boldsymbol{v}^*$ (solid lines, eq 26). The Kullback–Leibler divergence between male and female distributions is also reported. **A–B**: Male and female distributions are similar in shape as both sexes have the same demographic parameters (see left-hand side of panels for parameter values). However, females grow larger on average due to direct selection on size whereas males only track female size via selection for compatibility. **C–D**: Sex-specific demography generates asymmetries in size distributions with a higher frequency of small females. When sexual selection is strong enough, this leads to evolutionary branching (panel D) with one morph specialising on mating with smaller females. The smaller allele is seldom found at a homozygous state and the smaller large size mode is composed of heterozygotes. Other parameters: $c = 0.5$, $\mu_{e,f} = 0.1$, $K = 10^4$.

mate mostly with smaller females. In our simulations, such disruptive selection can be so strong that it overpowers directional selection before the population has completely converged to the singular strategy, leading to early branching (see Fig 4D). Under parameter values that favour a very wide distribution of female sizes, strong disruptive selection can also lead to the coexistence of more than two male morphs (see S2 Fig).

Evolutionary branching in our model hinges upon an advantage for males to specialize in different female sizes. Our results therefore connect to resource-competition and niche-partitioning models, where polymorphism evolves as individuals specialise to escape competition [e.g., 64–66], except that here the "resource" consists of potential mating partners. They also build on compatibility-based sexual selection models [56,57], which treat individual states as fixed traits determined at birth. In those models, male diversification arises in response to female genetic variation in compatibility traits generated by sexual conflict. By contrast, in our model females remain genetically monomorphic: variation in female states arises because individuals reproduce continuously while changing size through ontogeny. As a result, the population maintains a standing distribution of female sizes, which creates multiple mating niches; competition for these niches makes male fitness frequency dependent and can favour trait diversification. Another feature necessarily absent from these previous studies is that trait diversification here has knock-on effects on demography, altering the age and state structure of the population. In our model, the trait under sexual selection in males (investment into growth or sexual structures) also affects mortality via a survival–growth trade-off (eq. 23). Consequently, when selection favours alternative male growth strategies, evolutionary branching in that trait necessarily induces divergence in survival and lifespan. The resulting polymorphism therefore spans both mating strategy and life history, feeding back on population age structure and demography with the coexistence of fast-growing, short-lived males and slow-growing, long-lived males.

Alternative male mating morphs with markedly different life history traits are widespread (e.g., sneaker versus dominant males in fish, fighter versus sneaker males in beetles, and multiple morphs in isopods and ruffs). These polymorphisms are generally explained not by compatibility *per se*, but more often by sexual conflict (e.g., coercion versus resistance) or by condition-dependent development, where high- and low-condition males follow different trajectories [67,68]. While our results here apply mostly to the evolution of genetically determined alternative reproductive strategies [69], our model could readily be adapted to incorporate developmental plasticity and thus help disentangle the relative contributions of sexual selection, sexual conflict, and condition dependence in the evolution of sex-specific development. More generally, it provides a way to better understand the emergence and coexistence of alternative life-history strategies, together with their demographic causes and consequences.

**4.2.4 Extensions.** We extended the model to explore two cases. First, we investigated the sensitivity of our results to a change in ploidy by considering a haplodiploid system, where males are haploid and females are diploid. This boils down to adjusting the weights in the sums that define selection, using $\gamma_{fm} = 1$, $\gamma_{mm} = 0$, $q_f^\circ = 2(1-c)/(2-c)$, $q_m^\circ = c/(2-c)$, $\check{\nu}_f^\circ = \check{\nu}_m^\circ = 2c/3$ (see Box 2 and appendix A.5.2 in S1 Text), and $\partial z_m(u_m, v_m)/\partial u_m = 1$ (i.e., assuming complete dosage compensation). Because vital rates are defined as functions of individual phenotype and genetic effects are additive, haplodiploidy does not affect the singular trait values (S3 Fig). Male haploidy does however affect quadratic selection by fully exposing mutations in the male trait. By increasing $\partial z_m(u_m, v_m)/\partial u_m$, haploidy amplifies the male-specific component $h_{mm}(\boldsymbol{v}^*)$ of the Hessian matrix (see eq. 14). As a result, disruptive selection is stronger, especially under low male mortality and high sexual selection (S3D Fig). Evolutionary branching then produces the coexistence of two allelic types where the rarer, smaller type encodes larger male trait values than in the diploid case (compare S3D Fig and Fig 4D).

Second, we considered the effect of genetic constraints across sexes by assuming that males and females share the same growth rate (i.e., see eq 21). In a diploid population with additive gene action (i.e., $\partial z_f(u_f, v_f)/\partial u_f = \partial z_m(u_m, v_m)/\partial u_m = 1/2$), the age-specific directional selection coefficient on the shared trait $z_l(u_l, v_l)$ is the sum of the female and male gradients shown in eq (35):

$$\hat{s}_l(a, v_l) = \hat{s}_f(a, v_l) + \hat{s}_m(a, v_l), \tag{38}$$

resulting in sexual antagonism (i.e., when selection favours different trait values in males and females showing genetic correlations, [70,71]; Fig 2C, green line). Solving the integral of eq (38) for zero, we obtain a unique convergence-stable singular trait

$$v_l^* = \frac{1}{2} \frac{\int_0^\infty \left[(1-c)\lambda_f^x(a) + c\lambda_m^x(a)\right] da}{\int_0^\infty \left[(1-c)\lambda_f^l(a)l_f^o(a)\beta_f + c\lambda_m^l(a)l_m^o(a)\beta_m\right] da}.$$
(39)

Eq. (39) captures the lifetime trade-off between growth (numerator) and survival (denominator), averaged across sexes. In spite of this averaging, the equilibrium growth rate does not fall between the female and male optima found in the absence of genetic constraints (i.e., between $v_f^*$ and $v_m^*$ given by eq. 36) but instead evolves below both (Fig 3). This results from an evolutionary feedback: selection on males favours being smaller than females (owing to the costs of growth), genetic constraints then entail a reduction in female size, which in turn selects males to be even smaller, and so on. As a consequence of this sexually antagonistic feedback, females cannot evolve sufficiently large growth rates in response to high extrinsic mortality, making populations more prone to extinction than in the absence of genetic constraints (compare the green and red lines in Fig 3B).

Once the population has converged to $v_l^*$, whether or not polymorphism evolves depends on the age-specific quadratic selection coefficient $\hat{h}_{ll}(a, v_l^*) = \hat{h}_{ff}(a, v_l^*) + \hat{h}_{mm}(a, v_l^*)$, which is the sum of the female and male components (given by eq. 15; note that the mixed term $\hat{h}_{fm}(a, v_l^*)$ is always zero in our model, see eq. A-86). By computing $\hat{h}_{ll}(a, v_l^*)$ numerically, we find that genetic constraints across sexes prevent disruptive selection through two mechanisms. First, stabilizing selection on females is stronger, i.e., $\hat{h}_{ff}(a, v_l^*)$ is more negative at all ages, than when male and female traits evolve independently (S4 Fig, red lines). Sexual antagonism thus prevents disruptive selection from materialising in males because selection in females counteracts it. Second, because the equilibrium growth rate $v_l^*$ (eq 39) is lower owing to sexual antagonism, the variance in female size in the population is not large enough to generate disruptive selection on males, and for rare smaller males to gain an advantage (S5 Fig). This effect is especially pronounced under strong sexual selection (see S5D and S4 Figs).

## 5 Discussion

We derived different quantities for understanding selection on quantitative traits that influence fecundity, mortality, and development rates in age- and class-structured populations. First, the coefficients of directional selection (eqs 8–13) allow us to determine whether rare mutant alleles inducing small trait changes can invade and fix, and thus to identify convergence stable trait values toward which the population evolves. Second, the coefficients of quadratic selection (eqs 14–20) indicate whether such trait values are evolutionarily stable or prone to diversification via evolutionary branching, depending on whether nearby allelic types are eliminated by stabilizing selection or favoured by disruptive selection. Whether evolutionary branching occurs or not, these coefficients also identify the correlations among traits that are expected to be shaped by selection.

A key feature of our approach is that both directional and quadratic selection are expressed as a sum of age-specific allelic fitness effects. This provides a biologically meaningful partition of selective forces across ages. As shown in earlier single-class models [9,11], constructing such sums requires quantifying how trait expression at a given age influences both current reproduction (via fecundity at that age) and future reproduction (via changes in development and survival). Delayed effects are tracked using costates, which measure how changes in current internal states and survivorship affect future reproductive output (eq 11). In class-structured populations, these delayed effects must be characterised not only for each parental class but also with respect to each possible offspring class, requiring $n_c \times n_c$ systems of class-oriented costates (unless there are additional reproductive constraints, e.g., constant class proportions at birth; see Box 2). This reflects the biological reality that development can influence the production of different types of offspring in different ways, and that these effects can also depend on the parental class (e.g., sexes, ploidies, or habitats). By accounting for this, our results can be used to describe selection in systems where ontogeny interacts with reproductive choices (e.g., maternal effects on sex ratio; [72]), reproductive mode (e.g., allocation between sexual and vegetative reproduction; [73]), or dispersal (e.g., maternal effects on offspring dispersal; [74]).

With selection expressed in terms of age-specific fitness effects, we can consider and better understand the consequences of age-wise antagonistic selection, where different trait values are favoured at different ages but cannot evolve independently due to incomplete plasticity. This type of antagonism is an important aspect of life-history evolution and is often invoked in theories of senescence, because it enables the fixation of alleles that reduce survival later in life while providing advantages early on [13, 4]. Nevertheless, most continuous-time life-history models neglect this possibility by assuming complete age-wise plasticity in trait expression (e.g., [1–3,7,8,38,45,46,75]). By considering non-plastic traits and properly discounting selective effects at later ages [13], our model makes it possible to analyse the coevolution of ontogeny and senescence in the presence of genetic constraints. In our application for instance, intermediate longevity evolves because higher survival (i.e., via slower growth) is counter selected in early life, showing one possible way how age-wise antagonism may influence the evolution of life history.

By partitioning quadratic selection into age-specific components, our analysis also reveals cases of age-wise antagonistic quadratic selection, where stabilizing and disruptive selection act at different ages (or positive and negative correlational selection). In the sexual selection example, selection is stabilizing on younger males but disruptive on older ones because only the latter benefit from escaping intense competition for larger females (S4 Fig). More generally, our decomposition of quadratic selection shows that age-specific effects can be generated indirectly, i.e., through the cumulative influence of trait expression on internal state dynamics, survivorship dynamics, and class-structure (eqs 16b–16f). The dominant contribution to disruptive selection in the sexual selection example came from effects mediated by internal states, i.e., by size in this case (eq 37). Because developmental effects accumulate with age, while survivorship gradually declines, the resulting quadratic selection here is strongest at intermediate ages (S4 Fig). But in other systems different components may generate disruptive selection and thus follow different patterns of antagonism. Either way, the quadratic selection coefficients presented here can disentangle the mechanisms by which correlational and disruptive selection act through ontogeny and thereby help understand phenomena such as late differentiation, where morphs diverge only after a period of shared development [76], as well as other life-history polymorphisms where alternative reproductive tactics or morphologies emerge following common juvenile growth trajectories.

Our results also reveal how genetic constraints across classes can generate another form of antagonistic selection, when the same trait is expressed in multiple classes but selection favours different values in each. Such class-wise antagonism, particularly between sexes, has been argued to play an important role in shaping life-history evolution [77]. In primates, for example, strong sexual selection for increased size in males appears to cause a correlated increase in female size, owing to the shared genetic architecture of body size across sexes, despite being deleterious for females [78]. In our example model, sex-wise antagonistic selection prevents the diversification of male life histories, both because stabilizing selection on female growth counteracts disruptive selection on male growth, and because the resulting equilibrium size distribution is less conducive to disruptive selection. In future work, the derived selection coefficients could be used to study more systematically the role of such genetic constraints in shaping sex-specific ontogeny and alternative reproductive strategies, including cases where the underlying pleiotropic genes are expressed in classes with different ploidies (e.g., when traits are encoded on sex chromosomes). In fact, we have shown in our application that ploidy at the loci encoding a given trait can affect the likelihood to observe evolutionary branching, as well as the composition of the population after such event (see S3 Fig).

Antagonistic selection across ages and classes is expected to favour the evolution of plasticity, allowing traits to vary with context [79]. At first sight, our choice to model traits as fixed scalars may therefore seem restrictive, and even a step back compared to standard life-history models assuming complete plasticity (e.g., [1, 2]). However, modelling scalar traits does not mean that plasticity cannot be studied. Any dynamical system can in principle be represented by a finite recurrent neural network [80]. By treating the network's weights as evolving scalar traits (the vector $\boldsymbol{v}$ here), it becomes possible to generate reaction norms from constant genetic values in response to environmental or internal cues [81]. In this way, scalar-trait models can be extended to capture the complex mappings between gene expression to life-history

strategies, while integrating both external variables and internal state information (e.g., [82]), and these considerations apply to behavioral strategies alike [11]. Understanding how developmental and behavioral systems evolve to generate the required reaction norms, and thus studying how plasticity itself evolves gradually, remains an interesting challenge. By developing evolutionary models of scalar traits, including ontogeny and class structure, our results provide a step toward combining adaptive dynamics with systems approaches to better understand the long-term evolution of biological structures as integrated dynamical systems.

To conclude, we have decomposed directional and quadratic selection into age- and class-specific components, making it possible to identify the many trade-offs and antagonisms that can influence life-history evolution in structured populations. This provides a sharper understanding of adaptation in complex demographic settings and opens the door to a range of problems where context- or class-specific development plays a central role. These include the evolution of sex-specific ontogeny, state-dependent reproductive strategies, class-structured polymorphisms in plants and animals, dispersal strategies that depend on age or condition, and host–parasite interactions shaped by age-structure. More broadly, our approach contributes to linking the dynamics of development, demography, and adaptation, and to understanding how their interactions shape biological diversity.

## Supporting information

**S1 Fig. Male mortality and growth parameters influence male-specific evolution.** Each panel shows convergence stable growth rates $v^*$ (solid lines) and corresponding average sizes (dashed lines) for females (red) and males (blue), as a function of the parameter indicated on the $x$-axis (computed from eq 36 for trait values and B-99 for average sizes). Trait values under genetic constraint (i.e., identical growth rate in males and females, $v_l^*$) and the resulting female size are shown in green (from eq 39). **A**. Higher cost of growth to survival in males ($\beta_m$) selects for slower growth and smaller average size in males but does not affect females. **B**. Higher male extrinsic mortality ($\mu_{e,m}$) favours faster growth in males but does not affect females. Under genetic constraints, high values of $\mu_{e,m}$ also increase the frequency of females, tipping the balance in favour of even faster growth. Parameter values (unless varied on $x$-axis): $c = 0.5$, $\alpha = 0.2$, $\mu_{e,f} = \mu_{e,m} = 0.1$, $\beta_f = \beta_m = 0.2$, $\kappa = 1$ and $K = 10000$.
(TIFF)

**S2 Fig. Multiple branching and the coexistence of more than two male morphs.** The figure shows results from individual-based simulations (appendix B.3 in S1 Text) with independent growth rates and diploidy. The top part shows the expressed trait values in females (red) and males (blue): randomly sampled allelic values (dots; 20 copies every 10 time units) and population mean (solid lines). The bottom part shows male and female size distributions at equilibrium as histograms (averaged between time steps $10^5$ and $2 \times 10^5$). Parameter values: $c = 0.5$, $\alpha = 0.02$, $\mu_{e,f} = 0.1$, $\mu_{e,m} = 0.01$, $\beta_f = 0.1$, $\beta_m = 0.01$, $\kappa = 2$ and $K = 10000$.
(TIFF)

**S3 Fig. Evolutionary branching of male growth under haplodiploidy.** Each panel shows results from individual-based simulations (appendix B.3 in S1 Text). The top part shows the expressed trait values in females (red) and males (blue): randomly sampled allelic values (dots; 10 copies every 10 time units), population mean (solid lines), and analytically predicted convergence stable trait values (dashed lines). Reported values give: the male-specific component $h_{mm}^{xx}(v^*)$ (eq 14); its size-mediated component $h_{mm}^{xx}(v^*) = 0.25 \int_0^\infty \hat{h}_{mm}^{xx}(a, v^*) \, da$ (see eqs 37 and A-37, with 0.25 for diploidy); and the leading eigenvalue of $\mathbf{H}(v^*)$. As expected, when this eigenvalue is negative, the population remains monomorphic (panels A–C). When it is positive, evolutionary branching occurs in males (panel D, split in blue lines). The bottom part of each panel shows male and female size distributions at equilibrium as histograms (averaged between time steps $10^5$ and $2 \times 10^5$), along with the analytically predicted distribution at $v^*$ (solid lines, eq 26). The Kullback–Leibler divergence between male and female distributions is also reported. **A–B**: Male and female distributions are similar in shape as both

sexes have the same demographic parameters (see left-hand side of panels for parameter values). However, females grow larger on average due to direct selection on size whereas males only track female size via selection for compatibility. **C–D**: Sex-specific demography generates asymmetries in size distributions with a higher frequency of small females. When sexual selection is strong enough, this leads to evolutionary branching (panel D) with one morph specialising on mating with smaller females. Under haplodiploidy, net disruptive selection is stronger at the singular trait than in the diploid case, and the smaller male morph evolves smaller genetic values (compare values in Fig 4D and S3D Fig), but the distribution of male size at the evolutionary equilibrium is similar. Other parameters: $c = 0.5$, $\mu_{e,f} = 0.1$, $K = 10^4$.
(TIFF)

**S4 Fig. Age-specific quadratic selection.** Age-specific quadratic selection on growth (from eqs 35a and 35b) at the convergence stable strategies ($\mathbf{v}^* = (v_f^*, v_m^*) = (0.676, 0.589)$ for independent traits; $v_l^* = 0.219$ for a shared trait), under parameter values that produce disruptive selection on the male growth rate ($c = 0.5$, $\alpha = 0.1$, $\mu_{e,f} = 0.1$, $\mu_{e,m} = 0.05$, $\beta_f = 0.1$, $\beta_m = 0.01$, $\kappa = 2$ and $K = 10^4$, see Fig 4D). Plain lines correspond to the male and female age-specific components of disruptive selection under independent traits (given by eq 15; recall that here $\hat{h}_{jj} = \hat{h}_{jj,11}$). Dashed lines represent the same components, as well as their sum (green line) under genetic constraints (see eq. A-86).
(TIFF)

**S5 Fig. Genetic constraints between sexes prevent evolutionary branching.** Each panel shows results from individual-based simulations (appendix B.3 in S1 Text). The top part shows the expressed trait values in females and males (green): randomly sampled allelic values (dots; 10 copies every 10 time units), population mean (solid lines), and analytically predicted convergence stable trait values (dashed lines). Reported values give: the size-mediated component of quadratic selection $h_{ll}^{xx}(v_l^*) = 0.25 \int_0^\infty \hat{h}_{ll}^{xx}(a, \mathbf{v}^*)\mathrm{d}a$ (where $\hat{h}_{ll}^{xx}(a, v_l^*)$ is computed as in eq. A-86 using eq. A-37, with $0.25$ for diploidy); and the leading eigenvalue of $\mathbf{H}(v_l^*)$. Both are always negative showing size-mediated and net stabilising selection on growth, respectively. The bottom part of each panel shows male and female size distributions at equilibrium as histograms (averaged between time steps $10^5$ and $2 \times 10^5$), along with the analytically predicted distribution at $v_l^*$ (solid lines, eq 26). The Kullback–Leibler divergence between male and female distributions is also reported. **A–B**: Male and female distributions are the same since both sexes have the same growth rate and demographic parameters (see left-hand side of panels for parameter values). **C–D**: Sex-specific demography generates asymmetries in size distributions with a higher frequency of small females. Under genetic constraints, these differences are not sufficient to generate evolutionary branching. Other parameters: $c = 0.5$, $\mu_{e,f} = 0.1$ and $K = 10000$, and with simulated time steps of $0.2$ time units.
(TIFF)

**S1 Text. Appendices containing analytical results, supplementary results and a description of individual-based simulations.**
(PDF)

## Acknowledgments

We thank Dr. Ludovic Maisonneuve and two anonymous reviewers for constructive feedback.

## Author contributions

**Conceptualization:** Arthur Weyna, Charles Mullon, Laurent Lehmann.

**Formal analysis:** Arthur Weyna, Laurent Lehmann.

**Investigation:** Arthur Weyna, Charles Mullon, Laurent Lehmann.

**Visualization:** Arthur Weyna.

**Writing – original draft:** Arthur Weyna.

**Writing – review & editing:** Arthur Weyna, Charles Mullon, Laurent Lehmann.

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
