## [Decision Letter · Decision Letter 0]

28 Nov 2025

PCOMPBIOL-D-25-02008

Directional and disruptive selection in populations structured by class and continuous ontogeny under incomplete plasticity

PLOS Computational Biology

Dear Dr. Weyna,

Thank you for submitting your manuscript to PLOS Computational Biology. After careful consideration, we feel that it has merit but does not fully meet PLOS Computational Biology's publication criteria as it currently stands. Therefore, we invite you to submit a revised version of the manuscript that addresses the points raised during the review process.

We look forward to receiving your revised manuscript.

Kind regards,

Christian Hilbe

Academic Editor

PLOS Computational Biology

Natalia Komarova

Section Editor

PLOS Computational Biology

**Additional Editor Comments:**

Thank you for your submission. The paper is perhaps more on the technical end of the spectrum of papers typically sent to PLoS Computational Biology, but both reviewers very much appreciate the contribution.

Please take into account their very constructive comments on how to further improve the paper.

**Journal Requirements:**

We ask that a manuscript source file is provided at Revision. Please upload your manuscript file as a .doc, .docx, .rtf or .tex. If you are providing a .tex file, please upload it under the item type u2018LaTeX Source Fileu2019 and leave your .pdf version as the item type u2018Manuscriptu2019.

**Reviewers' comments:**

Reviewer's Responses to Questions

**Comments to the Authors:**

Reviewer #1: This study presents conditions for convergence stability and evolutionary branching in the context of trait systems in age- and class-structured populations, generalising previous theory. This seems like a solid contribution that is a necessary part of extending the basic theory of adaptive dynamics to more complex biological scenarios. The framework makes sense to me conceptually, but I am not qualified to check all of the mathematical details. My review is consequently restricted to more superficial aspects of the manuscript. Some general comments:

1. It was not always clear to me what restrictive assumptions are being made. For example, the set-up seems to exclude environmental effects (including indirect genetic effects) on the phenotype. It also seems to assume that development is deterministic. Some further guidance to the reader would help here.

2. More could be done to highlight the new insights from the example at the end, especially in the abstract.

3. Can evolutionary branching ever lead to the maintenance of more than two alleles in the example? Naively I would think that if selection for assortative mating is very strong, males should split up into even more morphs.

4. There is a substantial literature on size-assortative mating in various species of mollusc. Perhaps the authors could link their example to this literature.

More specific comments:

L44-60: Another point is that if plastic traits are governed by a large number of loci, each of which is only expressed in particular circumstances, then selection on each locus might be very weak.

L61, 88: ‘traits’

L116: It might be helpful to specify that n_x is independent of age.

Eq. 1: The survival probability function used in this equation is not introduced until two paragraphs later. Some signposting or rearrangement would be helpful.

Table 2: The notation v* for the singular strategy hasn’t been introduced yet. Maybe it could be added to the table legend.

L222: Does this actually correspond to any mathematical definition of curvature? I know that similar claims are made frequently in the theoretical literature on quadratic selection and certainly the second derivative is related to curvature. But most definitions of curvature have normalisation terms that depend on the first derivative and I see no guarantee that such terms will be negligible here.

L236-7: I don’t understand why mutation can’t move traits along a boundary.

L240: Mutational effects being exactly equal across traits is only one possible type of constraint on the mutational effects distribution, but it seems as if you treat this as the only form of such a constraint. See also L277 and L563 onwards.

Box 1: The co-states have no yet been defined. Please signpost.

P15, P26: The equation numbering does not continue correctly after the two boxes.

L630: Please specify that it is the per-male rate of egg production.

Fig. 3: This figure is quite hard to read, requiring switching back and forth between the legend and the figure. For example, finding the full set of parameter values for each panel requires reading both the top and side labels in the figure and the remaining parameters in the legend. Some redundancy and better signposting would be very helpful here.

Eq. 21: Please remove the comma at the end of the equation (grammatically the equation functions as the subject of the following sentence – the comma suggests instead that the equation functions as a sentence in its own right).

L707-8: Missing close parenthesis.

Reviewer #2: the review is uploaded as an attachment

**Have the authors made all data and (if applicable) computational code underlying the findings in their manuscript fully available?**

The PLOS Data policy requires authors to make all data and code underlying the findings described in their manuscript fully available without restriction, with rare exception (please refer to the Data Availability Statement in the manuscript PDF file). The data and code should be provided as part of the manuscript or its supporting information, or deposited to a public repository. For example, in addition to summary statistics, the data points behind means, medians and variance measures should be available. If there are restrictions on publicly sharing data or code —e.g. participant privacy or use of data from a third party—those must be specified.requires authors to make all data and code underlying the findings described in their manuscript fully available without restriction, with rare exception (please refer to the Data Availability Statement in the manuscript PDF file). The data and code should be provided as part of the manuscript or its supporting information, or deposited to a public repository. For example, in addition to summary statistics, the data points behind means, medians and variance measures should be available. If there are restrictions on publicly sharing data or code —e.g. participant privacy or use of data from a third party—those must be specified.

Reviewer #1: Yes

Reviewer #2: Yes

PLOS authors have the option to publish the peer review history of their article (what does this mean? ). If published, this will include your full peer review and any attached files.). If published, this will include your full peer review and any attached files.

**Do you want your identity to be public for this peer review?** For information about this choice, including consent withdrawal, please see our For information about this choice, including consent withdrawal, please see our Privacy Policy ..

Reviewer #1: No

Reviewer #2: No

**Figure resubmission:**
---

## [Editor Report · Decision Letter 1]

16 Feb 2026

Dear Dr. Weyna,

We are pleased to inform you that your manuscript 'Directional and disruptive selection in populations structured by class and continuous ontogeny under incomplete plasticity' has been provisionally accepted for publication in PLOS Computational Biology.

Best regards,

Christian Hilbe

Academic Editor

PLOS Computational Biology

Natalia Komarova

Section Editor

PLOS Computational Biology

Already the first time around, the two reviewers were very positive and suggested 'Accept' and 'Minor revisions', respectively.

The authors have carefully revised the manuscript, and they have addressed all remaining comments.

The paper is now suitable for publication in PLoS Computational Biology.

---

## [Editor Report · Acceptance letter]

PCOMPBIOL-D-25-02008R1

Directional and disruptive selection in populations structured by class and continuous ontogeny under incomplete plasticity

Dear Dr Weyna,

I am pleased to inform you that your manuscript has been formally accepted for publication in PLOS Computational Biology. Your manuscript is now with our production department and you will be notified of the publication date in due course.

With kind regards,

Zsofia Freund
